# Theoretical Foundations of Continual Learning via Drift-Plus-Penalty

**Nazreen Shah**                                                                 *nazreens@iiitd.ac.in*
*Dept. of Electronics and Communication Engineering,*
*Indraprastha Institute of Information Technology Delhi*
*New Delhi, India.*

**Govinda Arya**                                                                 *govindai@iiitd.ac.in*
*Dept. of Electronics and Communication Engineering,*
*Indraprastha Institute of Information Technology Delhi*
*New Delhi, India.*

**B. N. Bharath**                                                                 *bharathbn@iitdh.ac.in*
*Dept. of Electrical, Electronics and Communication Engineering,*
*Indian Institute of Technology Dharwad*
*Dharwad, India.*

**Ranjitha Prasad**                                                              *ranjitha@iiitd.ac.in*
*Dept. of Electronics and Communication Engineering,*
*Indraprastha Institute of Information Technology Delhi*
*New Delhi, India.*

**Reviewed on OpenReview:** *https://openreview.net/forum?id=QhxNMdhhBy*

## Abstract

In many real-world settings, data streams are inherently nonstationary and arrive sequentially, necessitating learning systems to adapt continuously without repeatedly retraining from scratch. Continual learning (CL) addresses this setting by seeking to incorporate new tasks while preventing catastrophic forgetting, whereby updates for recent data induce performance degradation on previously acquired knowledge. We introduce a control-theoretic perspective on CL that explicitly regulates the temporal evolution of forgetting, framing adaptation to new tasks as a controlled process subject to long-term stability constraints. We focus on replay-based CL settings in which a finite memory buffer preserves representative samples from prior tasks, allowing forgetting to be explicitly regulated. We propose COntinual Learning with Drift-Plus-Penalty (`COLD`), a novel continual learning framework based on the Drift-Plus-Penalty (DPP) principle from stochastic optimization. To facilitate theoretical and empirical analysis, we also consider an oracle variant, `COLD-ORACLE`, which serves as a reference benchmark. At each task, `COLD` and `COLD-ORACLE` minimize the instantaneous penalty corresponding to the current task loss while simultaneously maintaining a virtual queue that explicitly tracks deviations from long-term stability on previously learned tasks, hence capturing the stability–plasticity trade-off as a regulated dynamical process. We establish stability and convergence guarantees that characterize this trade-off, as governed by a tunable control parameter. Empirical results on standard benchmark datasets show that the proposed framework consistently achieves superior accuracy compared to a wide range of state-of-the-art CL baselines, while exhibiting competitive and tunable forgetting behavior that reflects the explicit regulation of the stability–plasticity trade-off through virtual queues and the DPP objective.

# 1   Introduction

Lifelong learning in artificial intelligence is motivated by the human capacity to continuously acquire and adapt knowledge over time. A central component of lifelong learning is continual learning (CL), which seeks to incorporate new information while retaining knowledge acquired from previous experiences. In streaming settings, memory constraints prevent the storage of all historical data, and only a limited subset of samples or summary statistics can be retained. These constraints make CL fundamentally more challenging than single-task learning, as joint training over the complete data from all tasks is infeasible. Conversely, sequentially updating a model using task-specific data often results in performance degradation on previously learned tasks, a phenomenon referred to as catastrophic forgetting McCloskey & Cohen (1989). Consequently, the objective of CL is to balance the acquisition of new knowledge (*plasticity*) with the preservation of previously learned information (*stability*).

The goal of CL mechanism is to regulate two competing objectives over time: the first objective is to achieve strong performance on the current task by effectively exploiting previously acquired knowledge, and the second objective is to control catastrophic forgetting by constraining the degradation of performance on earlier tasks as the model adapts to new data Lin et al. (2022). Task heterogeneity arising from shifts in data distributions across tasks acts as a hindrance that induces conflicting update directions and as a result, optimizing current-task performance and enforcing stability on past tasks constitute a coupled regulation problem with an inherent trade-off. Existing strategies for CL fall into three categories: regularization-based Kirkpatrick et al. (2017); Li & Hoiem (2017); Liu & Liu (2022), replay-based Lopez-Paz & Ranzato (2017); Chaudhry et al. (2018); Caccia et al. (2021), and architecture-based methods Rusu et al. (2016); Long et al. (2025). Regularization- and Architecture-based CL methods primarily mitigate forgetting in a constraint manner through capacity allocation or parameter-level penalties, respectively, but do not explicitly model the temporal evolution of forgetting or provide principled mechanisms to regulate long-term stability under task heterogeneity.

Replay-based CL methods provide an explicit mechanism to observe and quantify forgetting on previously learned tasks through budget-constrained memory samples. Notable techniques that employ memory replay include Gradient Episodic Memory (GEM) Lopez-Paz & Ranzato (2017), Averaged GEM (`A-GEM`) Chaudhry et al. (2018), and Non-Convex CL with Adaptive Learning Rates (`NCCL`) Han et al. (2023). Approaches such as GEM and `A-GEM` enforce per-step feasibility by projecting the current task gradient onto a constraint set defined by past-task gradients, thereby suppressing updates that conflict with previously learned tasks. Although such projection-based strategies prevent instantaneous interference, they do not control the accumulation of forgetting over time and may substantially reduce plasticity under high task diversity, limiting adaptation to new tasks. `NCCL` formulates CL as a nonconvex stochastic optimization problem with replay, but it lacks mechanisms to ensure long-term stability with respect to catastrophic forgetting.

We argue that CL is inherently a dynamic process and not a static optimization problem, as the learner must adapt to new tasks while constraining the progressive degradation of previously acquired knowledge. Moreover, catastrophic forgetting evolves and accumulates over successive updates. Hence, long-term stability of the learning dynamics is as important as instantaneous task-level performance. This temporal coupling motivates a control-theoretic formulation of CL, instead of a conventional optimization-based formulation. Accordingly, we adopt tools from stochastic control, in particular Lyapunov analysis and the drift-plus-penalty (DPP) framework Neely (2006), to explicitly model and regulate the dynamics of forgetting under resource constraints. The DPP framework introduces a virtual queue that tracks cumulative constraint violations over time, transforming CL into a problem of stabilizing this queue while optimizing task performance. At each step, the learner selects actions that minimize a weighted combination of the immediate learning objective and the Lyapunov drift of the virtual queue, i.e., balancing short-term plasticity with long-term stability. This dual-objective strategy renders DPP as an important approach for CL, as it provides a natural mechanism to *control* the stability–plasticity trade-off.

**Contributions:** We cast CL as a long-term constrained stochastic control problem, where catastrophic forgetting is modeled as the accumulation of constraint violations over time. Specifically, we propose *COntinual Learning via Drift Plus Penalty (COLD)*, a principled stochastic control framework for continual learning based

on Lyapunov optimization. We additionally introduce `COLD-ORACLE`, an oracle benchmark used for analysis and performance evaluation. The key contributions of this work are as follows:

- **Explicit stability–plasticity trade-off:** We formalize CL as a long-term constrained optimization problem and show that both algorithms admit an explicit and tunable trade-off between performance on the current task and forgetting of past tasks through a single parameter $V$. As in standard Lyapunov optimization, our analysis too establishes an $O(1/V)$ performance guarantee versus an $O(V)$ forgetting bound, providing a clear and interpretable characterization of the stability–plasticity trade-off that is absent in existing methods in CL.

- **Performance guarantees dependent on task variation:** Unlike prior CL analyses, which are agnostic to how tasks evolve over time, our theoretical bounds explicitly depend on task variation measures that quantify changes in loss functions across tasks. This reveals, for the first time, how large inter-task variability fundamentally limits CL performance, thereby connecting algorithmic behavior to problem nonstationarity.

- **Projection-free constrained learning with fixed learning rates:** The proposed method avoids gradient projection or dual-update mechanisms commonly used in constrained CL methods such as GEM, A-GEM, and related approaches. Instead, constraints are enforced implicitly through virtual queue dynamics, enabling projection-free updates and preserving plasticity.

- **Queue-based quantification of forgetting:** We introduce a virtual-queue mechanism to dynamically track cumulative forgetting across tasks and establish queue stability guarantees. This provides a rigorous, algorithm-dependent notion of forgetting that complements performance bounds and offers deeper insight than retrospective or heuristic forgetting metrics used in prior work.

**Novelty:** The proposed framework is a principled stochastic control perspective for CL by adapting the DPP framework. Unlike previous approaches that approximate stability Farajtabar et al. (2020) or rely on heuristic surrogate constraints Chaudhry et al. (2018) to manage forgetting, `COLD` and `COLD-ORACLE` integrate performance and stability objectives into a single unified optimization objective. DPP allows for analytical control of the stability–plasticity trade-off, based on virtual queue-based dynamics to track constraint violations. By modeling memory interference as long-term constraint violations, our algorithms enjoy good theoretical guarantees on convergence and bounded forgetting via theoretical analysis on queue stability. This is an advancement over existing methods that typically lack theoretical guarantees. Moreover, by avoiding projection-based updates as in Lopez-Paz & Ranzato (2017); Chaudhry et al. (2018), proposed methods circumvent the loss of plasticity and suboptimal task adaptation that can arise when gradients are forcibly restricted to feasibility regions defined by past tasks. More recent formulations, such as that of Lin et al. (2023), control forgetting only in expectation and do not provide an explicit mechanism to regulate constraint violations over time. In contrast, our algorithms offer a *controllable* and *algorithm-dependent* framework that operates without projections, applies to non-convex objectives, and provides explicit guarantees on long-term forgetting via queue stability. To the best of our knowledge, ours is the first method to adapt DPP for memory-based CL with quantified trade-offs between stability and plasticity.

**Technical challenges:** It is important to note that our setting introduces several nontrivial technical challenges that necessitate substantial departures from standard arguments in DPP analysis. Classical DPP formulations assume independent state evolution, which allows performance to be benchmarked against a stationary solution. In contrast, the constraints in our problem are endogenous and trajectory-dependent: the constraint functions depend explicitly on the learner's own past iterates rather than on exogenous, independent stochastic processes. This dependence precludes direct comparison with a stationary benchmark and instead requires the introduction of an idealized CL problem as a reference. Bounding the average queue length relative to this new benchmark is itself nontrivial. Moreover, the learning environment is inherently nonstationary, with task-dependent loss functions that may vary arbitrarily over time, rendering steady-state or ergodic arguments which are central to classical DPP analysis, inapplicable. Consequently, both the performance bound and the average queue stability bound must be expressed in terms of task variation. Finally, establishing a bound on the average gradient via the gradient descent updates applied to the DPP term presents additional difficulties and has no direct analogue in the classical DPP framework.

Taken together, these challenges make the analysis substantially more delicate than a straightforward reuse of existing DPP results and require the development of new bounds that explicitly couple task variation, queue dynamics, and optimization error.

## 2 Related Works

**Rehearsal/Replay-Based Methods:** Rehearsal-based methods employ a limited memory buffer of past samples, which is replayed alongside incoming data to mitigate catastrophic forgetting. Experience Replay (ER) serves as the foundational rehearsal-based baseline in CL by maintaining a fixed-size episodic memory of samples from previously encountered tasks and jointly training on mini-batches composed of both current and buffered data Rolnick et al. (2019). Extensions of ER such as Dark Experience Replay (`DER`/`DER++`) enhance stability by storing and distilling past model outputs Buzzega et al. (2020), Meta-Experience Replay (`MER`) incorporates meta-learning principles to reduce gradient interference across tasks Riemer et al. (2019), and `ER-ACE`, which introduces asymmetric losses to control representation drift. Among other works, `GDumb` Prabhu et al. (2020) adopts a simple strategy by greedily populating the buffer and retraining a model from scratch at evaluation time, `CBA` Wang et al. (2023) adapts classifier bias to mitigate recency effects in online CL, and `REFRESH` Wang et al. (2024) proposes a mechanism to improve performance by integrating an unlearn-then-relearn strategy. Recent work also advances rehearsal through adaptive cognitive allocation in buffer construction Zhang et al. (2024) alongside addressing plasticity-stability trade-off. While rehearsal-based methods mitigate forgetting by repeatedly replaying stored samples or representations to implicitly regularize parameter updates, they do not explicitly model or regulate the temporal accumulation of forgetting. In the federated setting, Keshri et al. (2025) propose a replay-memory–based continual federated learning strategy (C-FLAG) to global catastrophic forgetting and also provides convergence guarantees.

**Drift Plus Penalty in Machine Learning:** The DPP technique, first introduced in the context of stochastic network optimization Neely (2010), is a powerful framework to design stochastic control algorithms that achieve long-term performance guarantees under uncertainty. DPP is based on Lyapunov optimization that balances two competing objectives: minimizing a time-averaged penalty and ensuring constraint satisfaction through the control of virtual queues that are based on time-averaged constraint violations. The DPP framework has found broad applications across domains such as wireless resource allocation Neely et al. (2010), energy-efficient communications Samarakoon et al. (2015), and dynamic scheduling, where it ensures online decision-making with provable convergence bounds. Recently, DPP has been used for constrained reinforcement learning, where the algorithm balances reward maximization with long-term constraint satisfaction Huang et al. (2021); Xu et al. (2025). In online learning settings, it enables dynamic regularization and adaptive constraint enforcement, leading to the best known static regret bounds Sinha & Vaze (2024); Sarkar et al. (2025); Vaze & Sinha (2025).

**CL via Constrained Optimization:** CL is often cast as a constrained optimization problem, where the goal is to minimize the loss on the current task while enforcing constraints on forgetting. Classic regularization-based methods such as `EWC` Kirkpatrick et al. (2017) and Memory-Aware Synapses (MAS) Aljundi et al. (2018) penalize changes to parameters that are critical for previous tasks, but they do not dynamically track forgetting. Memory replay-based methods with gradient-level constraints include GEM Lopez-Paz & Ranzato (2017) and `A-GEM` Chaudhry et al. (2018); they impose constraints through gradient projection so that the loss on the previous tasks does not increase. `NCCL` adapts task-specific learning rates to better balance stability and plasticity Han et al. (2023). In effect, `GEM`, `A-GEM`, and `NCCL` projects the current task gradients onto a feasible region defined by past-task gradients, which results in components of the original update direction being discarded, especially those that conflict with past-task directions. In cases of high task diversity, a projection operation can lead to reduced plasticity, limiting the ability to fully adapt to new tasks. More recently, techniques for online CL have been proposed for learning from non-stationary data, where each sample can be seen only once Urettini & Carta (2025); Wu et al. (2024). While online CL is necessary in streaming settings, it severely limits the learner's ability to mitigate catastrophic forgetting, since past data cannot be revisited for corrective updates. Existing online methods are largely local and per-step, lacking principled mechanisms to control forgetting accumulation and to provide long-term stability guarantees under heterogeneous task distributions.

The remainder of the paper is organized as follows. In section 3, we introduce the idealized CL formulation (see (2)) and propose a relaxed version in (3). Later, we derive the corresponding DPP-based constrained optimization framework. Section 4 develops a more practical and implementable approximation of this formulation, leading to the proposed `COLD` framework, and the corresponding theoretical analysis is done in section 5, including queue stability and trade-off guarantees. Section 6 proposes a gradient based optimization for the proposed algorithms, and present the corresponding guarantees. Finally, Section 7 provides experimental evaluation and comparisons with existing CL methods.

## 3 Continual Learning: Problem Formulation

We consider the CL problem where the challenge is to learn from the tasks that arrive sequentially over time, i.e., the learning dynamics must be explicitly controlled to ensure progress on current tasks while constraining cumulative degradation on past tasks. Consider a sequence of tasks denoted as $\mathbb{T} := \{1, 2, \ldots, T\}$,[1] where each task $t \in \mathbb{T}$ is associated with a data set $\mathcal{D}_t := \{(\mathbf{x}_{t,i}, y_{t,i}) : i = 1, 2, \ldots, n_t\}$ drawn from a distribution $P_t$. Here, $\mathbf{x}_{t,i} \in \mathcal{X}$ is the feature vector and $y_{t,i} \in \mathcal{Y}$ is the corresponding target label. The quality of the predictor $h_{\mathbf{w}}(\mathbf{x}) \in \mathcal{Y}$ with the model $\mathbf{w} \in \mathcal{W} \subseteq \mathbb{R}^d$ for an input feature vector $\mathbf{x}$ from any task $t \in \mathbb{T}$ is evaluated using an average loss function $\Phi_t$ defined by

$$\Phi_t(\mathbf{w}) := \frac{1}{n_t} \sum_{i=1}^{n_t} l(h_{\mathbf{w}}(\mathbf{x}_{t,i}), y_{t,i}), \tag{1}$$

where $l : \mathcal{Y} \times \mathcal{Y} \to \mathbb{R}^+$ is the loss function (eg: cross-entropy loss). The goal of CL after observing the data of task $t \in \mathbb{T}$ is to learn a model $\mathbf{w}$ that performs well for all tasks $\tau = 1, 2 \ldots, t$.

A straightforward solution to the problem stated above is to optimize jointly with respect to all tasks, i.e., solve a multitask problem: $\min_{\mathbf{w}} \sum_{\tau=1}^{T} \Phi_\tau(\mathbf{w})$. The following are some roadblocks to using the multitask approach:

- Multitask approach requires access to all the tasks' data points. However, the memory requirement scales with the number of tasks $T$, and therefore becomes impractical.

- The computational complexity scales with the number of tasks.

- The data distribution for future tasks is unknown.

A naive approach to overcome the above is to minimize the loss incurred in the current task, i.e., find $\mathbf{u}_t \in \arg\min_{\mathbf{w}} \Phi_t(\mathbf{w})$, where $\mathbf{u}_t$ denotes a solution that is optimal only for the current task and ignores all past-task constraints. This leads to a phenomenon called the *catastrophic forgetting* French (1999), i.e., the loss $\Phi_\tau(\mathbf{u}_t)$ can be very large for one or more tasks with task index $\tau \in \{1, 2, \ldots, t-1\}$.

**Problem Setting:** To alleviate catastrophic forgetting under storage constraints, we assume access to a finite memory buffer that retains a limited number of representative samples from each previously observed task. We assume that the total memory budget is $M$ samples. This implies that an $m = \lfloor \frac{M}{T} \rfloor$ amount of memory is allocated to each task. In practice, $m$ points are uniformly randomly sampled from the data points of each task, i.e., the probability that a subset of samples of size $m$ is chosen from the task $t$ is $\frac{1}{\binom{n_t}{m}}$. We use $\hat{\Phi}_\tau(\mathbf{w})$ to denote the empirical loss incurred in the task $\tau$ using $m$ memory samples stored by sampling task $\tau$. We first present an *ideal* CL problem in which per-task forgetting is controlled relative to the task-wise optimal loss, yielding an algorithm-independent benchmark, as follows:

$$\begin{aligned} \min_{\mathbf{w}} \quad & \Phi_t(\mathbf{w}) \\ \text{subject to} \quad & \hat{\Phi}_k(\mathbf{w}) - \inf_{\mathbf{v} \in \mathcal{W}} \hat{\Phi}_k(\mathbf{v}) \leq \delta', \text{ for all } k < t, \end{aligned} \tag{2}$$

---

[1]Task $i$ may correspond to some task denoted $\mathcal{T}_i$. An example scenario is that the image data sets are revealed with new tasks each time. The task could be identifying, say, a cat in the first task while in the second task a dog, and so on.

for $\delta' > 0$. In general, a very small value of $\delta'$ may turn the problem infeasible, as the subsequent task may differ significantly from the current task $t$. Thus, the feasibility of (2) depends jointly on task similarity, model capacity, and memory-induced estimation error; throughout, $\delta'$ is treated as an intrinsic problem parameter capturing these effects. Here, the constraints on forgetting are enforced empirically using the stored memory, which is the only accessible proxy for data from the past tasks. The above problem in general may not be feasible for any value of $\delta'$. For example, when the loss $\Phi_t(\mathbf{w})$ or $\hat{\Phi}_t(\mathbf{w})$ changes arbitrarily across task $t$, one may need a large $\delta'$ to accommodate large variation of losses. On the other hand, when the losses do not change, then a fixed small $\delta'$ makes the problem feasible. Therefore, the *smallest feasible* $\delta'$ implicitly reflects the degree of regularity in how the losses of each task vary. [2] Assuming feasible $\delta'$, let $\mathbf{w}_t^*$ be the optimal solution to the above problem at task $t$. While the above problem is ideal, it cannot be implemented in real time since the optimal solutions in general are inaccessible in a CL setting. To overcome this, consider the following problem where at task $t$ we aim to minimize the *average* loss while limiting the *average* forgetting:

$$\min_{\mathbf{w}_1,\ldots,\mathbf{w}_t} \quad \frac{1}{t} \sum_{\tau=1}^{t} \Phi_\tau(\mathbf{w}_\tau)$$

$$\text{subject to} \quad \frac{1}{t-1} \sum_{k=1}^{t-1} \left( \hat{\Phi}_k(\mathbf{w}_t) - \hat{\Phi}_k(\tilde{\mathbf{w}}_{t,k}) \right) \leq \delta, \text{ for a given } \delta, \tag{3}$$

where a reference model $\tilde{\mathbf{w}}_{t,k}$ is used to benchmark the performance of the current model $\mathbf{w}_t$ at task $t$ on the past task $k$. This model is drawn from the trajectory of an actual algorithm instead of an unattainable optimal model, thereby making the constraints operational under memory and sequential-access limitations. The tolerance $\delta$ must exceed $\delta'$ to absorb (i) suboptimality of reference models and (ii) averaging-induced relaxation. Consequently, the constraint in (3) controls average cumulative forgetting over time rather than enforcing strict per-task feasibility at each step, and the tolerance parameter $\delta$ absorbs both the suboptimality of the reference models and temporal aggregation. Average constraints prevent a single outlier task from dominating updates while still guaranteeing asymptotic feasibility via queue stability. In this sense, (3) serves as a tractable relaxation of (2). This formulation shifts from an oracle benchmark to an algorithm-dependent one, enabling long-term stability guarantees that can be analyzed and enforced. In particular, we use (3) to design an algorithm, and provide guarantees in relation with the problem in (2). Further, since the reference model is generally suboptimal for individual tasks compared to $\min_{\mathbf{v}} \hat{\Phi}_k(\mathbf{v})$, we use a different constraint $\delta$ instead of $\delta'$ in (2). As opposed to classical online learning or stochastic optimization problems, the constraint set is increasing with the number of tasks. We assume that an algorithm $\mathcal{A}$ aimed at solving (3) at task $t$ results in a sequence of solutions $\mathbf{w}_1, \ldots, \mathbf{w}_{t-1}$. In this paper, we consider the following two reference models with respect to an algorithm $\mathcal{A}$:

1. *Case* 1*:* If we want the current model to perform better than the previously obtained model $\mathbf{w}_{t-1}$ from $\mathcal{A}$, then our reference model becomes $\tilde{\mathbf{w}}_{t,k} = \mathbf{w}_{t-1}$ for $k < t$ Lopez-Paz & Ranzato (2017).

2. *Case* 2*:* The second choice of reference model at task $t$ is the following

$$\tilde{\mathbf{w}}_{t,k} := \operatorname*{arg\,min}_{\mathbf{w} \in \{\mathbf{w}_1,\ldots,\mathbf{w}_{t-1}\}} \hat{\Phi}_k(\mathbf{w}), \tag{4}$$

where the minimization is with respect to all the models returned by algorithm $\mathcal{A}$. Here, $k = 1, 2, \ldots, t-1$. It is important to note that $\hat{\Phi}_k(\tilde{\mathbf{w}}_{t,k}) \leq \hat{\Phi}_k(\mathbf{w}_k)$ for all $k$. In practice, Case 2 can be implemented using a rolling window or compressed model summaries; our analysis applies to any subset containing the empirical minimizer.

Existing memory-based approaches such as GEM and `A-GEM` rely on approximating the problem given in (3) (case 1 with $\delta = 0$) by a quadratic optimization problem, resulting in algorithms based on projections. This leads to sub-optimal performance on the current task, particularly in high-diversity cases where the true gradient lies outside the feasible region defined by memory gradients. To the best of our knowledge, `COLD` is the first-of-its-kind projection-free method to leverage a control-theoretic framework in CL that offers a principled way to quantify the balance in stability and plasticity.

---

[2]In a later section, we show that under some regularity conditions, there always exists a $\delta' > 0$ that results in a feasible solution to the above problem.

## 4 Proposed Algorithm

To accommodate the constraints in the optimization problem formulated in (3), we adopt the Lyapunov optimization approach, a technique commonly employed in the analysis and control of queuing systems Neely (2013). In this approach, the per-task objective is to minimize a weighted combination of the average loss on the current task and a drift term that captures constraint violations. To quantify the drift, we maintain a set of $t-1$ virtual queues after observing task $t \in \mathbb{T}$, which are updated as follows:

$$Q_k[t] = \max \left\{ Q_k[t-1] + \Delta \hat{\Phi}_k(\mathbf{w}, \tilde{\mathbf{w}}_{t,k}), 0 \right\}, \tag{5}$$

where $\Delta \hat{\Phi}_k(\mathbf{w}, \tilde{\mathbf{w}}_{t,k}) := \hat{\Phi}_k(\mathbf{w}) - \hat{\Phi}_k(\tilde{\mathbf{w}}_{t,k}) - \delta$, $k = 1, 2, \ldots, t-1$. Recall that $\tilde{\mathbf{w}}_{t,k}$ is the reference model obtained by our proposed algorithm. Further, the queues are initialized with $Q_k[0] = 0$ for all $k$. Here, $\mathbf{w}$ denotes the model output by the algorithm upon completing the task $t$ that needs to be optimized. Each queue $Q_k[t]$ acts similar to that of a Lagrange multiplier that adaptively penalizes forgetting on task $k$. Lagrange multipliers are typically updated using dual descent, while the queue updates are simple and cheaper. These virtual queues track the degree of constraint violation where a large value of $Q_k[t]$ for some $k < t$ indicates significant forgetting on the task $k$. Therefore, keeping these queues stable by ensuring that their values remain small leads to the mitigation of catastrophic forgetting and promotes satisfaction of the average constraint over the tasks. Towards analyzing this, consider the following Lyapunov function

$$\mathcal{L}_{\mathbf{w}_t}[t] := \frac{1}{2(t-1)} \sum_{k=1}^{t-1} Q_k^2[t], \tag{6}$$

where $\mathbf{w}_t$ is the model obtained by algorithm $\mathcal{A}$ at task $t$. The Lyapunov function encapsulates all the constraints into one function. We adapt the following intial condition: $\mathcal{L}_{\mathbf{w}_0}[0] = \mathcal{L}_{\mathbf{w}_1}[1] = 0$. The Lyapunov drift at task $t$ can be written as Neely & Huang (2010)

$$\Delta \mathcal{L}[t] := \mathcal{L}_{\mathbf{w}_t}[t] - \mathcal{L}_{\mathbf{w}_{t-1}}[t-1]. \tag{7}$$

The DPP algorithm minimizes a weighted sum of the current task loss and the Lyapunov drift, thereby balancing long-term constraint satisfaction with immediate performance. Mathematically, for each task $t$ we have

$$\texttt{DPP-Problem:} \inf_{\mathbf{w}} \left[ V \Phi_t(\mathbf{w}) + \Delta \mathcal{L}[t] \right], \tag{8}$$

where $V > 0$ is the parameter of the algorithm that trades off the current loss and the drift from the learning from previous tasks (forgetting). Unfortunately, the drift term in the above optimization problem is difficult to handle. An approach followed in the literature is to use an upper bound on the drift as derived in the following lemma.

**Lemma 1.** *The drift at task $t$ using model $\mathbf{w}$ is bounded as*

$$\Delta \mathcal{L}[t] \le \Delta_t(\mathbf{w}, \mathbf{w}_{t,ref}) + \sum_{k=1}^{t-1} Q_k[t-1] \Delta \hat{\Phi}_k(\mathbf{w}, \tilde{\mathbf{w}}_{t,k}),$$

*where $\Delta_t(\mathbf{w}, \mathbf{w}_{t,ref}) := \frac{1}{2(t-1)} \sum_{k=1}^{t-1} (\Delta \hat{\Phi}_k(\mathbf{w}, \tilde{\mathbf{w}}_{t,k}))^2$, $\Delta \hat{\Phi}_k(\mathbf{w}, \tilde{\mathbf{w}}_{t,k}) := \hat{\Phi}_k(\mathbf{w}) - \hat{\Phi}_k(\tilde{\mathbf{w}}_{t,k}) - \delta$, and $\mathbf{w}_{t,ref} := (\tilde{\mathbf{w}}_{t,1}, \ldots, \tilde{\mathbf{w}}_{t,t-1})$ is the vector of reference models.*

*Proof:* See Appendix.

The second term in the expression captures the extent to which the constraints are violated as captured by $Q_k[t-1]$ term. The queue acts as a multiplicative constant to $\Delta_t(\mathbf{w}, \mathbf{w}_{t,\text{ref}})$, which captures the task $k$ violation in the current task $t$. Thus, larger second term enforces us to optimize the current model $\mathbf{w}$ to handle the violation of task $k$, enabling stability.Unlike the classical DPP approach, we have retained the

first term in its data-dependent form, $\Delta_t(\mathbf{w}, \mathbf{w}_{t,\text{ref}})$ which directly reflects the empirical variation induced by the current model relative to the reference models. This refinement yields a tighter characterization of the drift, allowing our theoretical guarantees to provide a better insight into the effect of variation of the models across tasks. In the proposed algorithm, we solve the following problem

$$\mathbf{w}_t \in \arg\inf_{\mathbf{w}} \left\{ \text{DPP}_{V,t}(\mathbf{w}) := V\Phi_t(\mathbf{w}) + \sum_{k=1}^{t-1} Q_k[t-1]\Delta\hat{\Phi}_k(\mathbf{w}, \tilde{\mathbf{w}}_{t,k}) \right\}. \tag{9}$$

Our method is presented in Algorithm 1.

---

**Algorithm 1 CO**ntinual **L**earning **D**PP (`COLD`)/(`COLD-ORACLE`)

---

1: **Initialize:** $Q_1[0] = 0$, random initialization of $\mathbf{w}_0$, $\delta \geq 0$, and $V > 0$.
2: **for** $t = 1, 2, \ldots, T$ **do**
3:     Receive data of task $t$.
4:     Sample a batch $\mathcal{B}_{t-1}$ from previous tasks with $|\mathcal{B}_{t-1}| = m$.
5:     Obtain a reference model:
6:         – Case 1: (`COLD`) $\tilde{\mathbf{w}}_{t,k} = \mathbf{w}_{t-1}$ or
7:         – Case 2: (`COLD-ORACLE`) $\tilde{\mathbf{w}}_{t,k}$ by solving (4).
8:     **DPP step:** Solve (9) to obtain $\mathbf{w}_t$.
9: **end for**

---

The proposed algorithm is shown in Algorithm 1. If the algorithm uses Case 1, then the memory requirement in terms of storing the model does not scale with $T$ as it only needs to store the immediate past model of size $d$, making it efficient. We term this as `COLD`. On the other hand, Case 2 requires storage of all the past models to solve (4) leading to a linear scaling of storage complexity, and we term the corresponding algorithm `COLD-ORACLE`. Although, `COLD-ORACLE` complexity scales linearly with $T$, this can be mitigated via task sub-sampling, windowed replay, or compressed model summaries. The study of such techniques is relegated to future work. The proposed method in Algorithm 1 requires a solution to (9); a stationary solution can obtained by using Gradient Descent (GD) approach. Our first set of theoretical results assumes an optimal solution for (9). Later, we extend our results to a more practical GD approach.

## 5 Performance Analysis of the `COLD`/`COLD-ORACLE` Algorithm

In this section, we present first set of theoretical results. We make the following standard assumptions on the loss function. The proofs of all theorems and lemmas can be found in the appendix.

**Assumption 1.** *(Smoothness) The loss functions $\Phi_t(\mathbf{w})$ and $\hat{\Phi}_t(\mathbf{w})$ are assumed to be $L$ and $l$ smooth, respectively. Mathematically, $\forall\, \mathbf{w}, \mathbf{u} \in \mathcal{W}$ and $\forall t$*

$$\Phi_t(\mathbf{w}) \leq \Phi_t(\mathbf{u}) + \langle \mathbf{w} - \mathbf{u}, \nabla\Phi_t(\mathbf{u}) \rangle + \frac{L}{2}||\mathbf{w} - \mathbf{u}||^2, \tag{10}$$

*and $L$ replaced by $l$ for $\hat{\Phi}_t(\mathbf{w})$.*

In Step 8 of Algorithm 1, we assume that the optimization problem is solved exactly, yielding an optimal solution denoted by $\mathbf{w}_t$. Next, we present the main results of the paper that characterizes the optimality gap and the average queue size.

**Theorem 1.** *Given Assumption 1, the `COLD`/`COLD-ORACLE` algorithm (with Case 1 or Case 2) leads to the following performance bound*

$$\frac{1}{T}\sum_{t=1}^{T}[\Phi_t(\mathbf{w}_t) - \Phi_t(\mathbf{w}_t^*)] \quad \leq \quad \frac{1}{VT}\sum_{t=1}^{T}\Delta_t(\mathbf{w}_t, \mathbf{w}_{t,ref}), \tag{11}$$

*where $\mathbf{w}_t^*$ is an optimal solution of (2) with $\delta' < \delta$.*

The left-hand side above corresponds to the loss difference between the proposed algorithm and the optimal solution obtained by solving (2)–the ideal CL problem. The above upper bound captures the effectiveness of the algorithm. Note that the constraint is governed by the difference between the model obtained by the algorithm and the reference model, i.e., $\mathbf{w}_t$ and $\tilde{\mathbf{w}}_{t,k}$, $t \in \mathbb{T}$. The latter is captured through $\Delta_t(\mathbf{v}, \mathbf{u})$, as defined in Lemma 1. This appears in the the upper bound, which reflects the algorithm-dependent error arising from constraint handling, and hence, it can be evaluated using the iterates $\{\mathbf{w}_t\}$ produced by the COLD/COLD-ORACLE algorithm. Further, it is easy to show that if $\Delta_t(\mathbf{w}_t, \mathbf{w}_{t,\text{ref}})$ is bounded (well regularized algorithmic updates), then $\frac{1}{T} \sum_{t=1}^{T} \Delta_t(\mathbf{w}_t, \mathbf{w}_{t,\text{ref}})$ is bounded, and hence the upper scales as $\mathcal{O}(1/V)$. Therefore, if $V$ is large, then the upper bound reduces, leading to a better performance guarantee, as expected. However, we expect that larger $V$ results in higher forgetting, leading to a trade-off. This is the essence of our next result on the constraint quantification (or queue stability analysis as in stochastic optimization). Towards stating the result, we need the following definition.

**Definition 1.** *Let $f_t(\mathbf{w})$ and $g_t(\mathbf{w})$, $t = 1, 2, \ldots, T$ be two sequences of functions. The loss variation at $t$ is defined as follows:*

$$D_{f,g}[T] := \frac{1}{T} \sum_{t=1}^{T} \sup_{\mathbf{w}} |f_t(\mathbf{w}) - g_{t-1}(\mathbf{w})|. \tag{12}$$

*Similarly, for a fixed sequence $f_t(\mathbf{w})$, we define $D_f[T] := \frac{1}{T} \sum_{t=1}^{T} \sup_{\mathbf{w}} |f_t(\mathbf{w}) - f_{t-1}(\mathbf{w})|$.*

The above definition is used to capture how the loss functions change, as in $D_{f,g}[T]$ or between two consecutive tasks as in $D_f[T]$, uniformly over the parameter space. While the above definition uses a uniform bound for analytical clarity, in practice it suffices that the deviation is bounded over the sequence $\{\mathbf{w}_t\}$. Next, we present a bound on the average queue size.

**Theorem 2.** *Given Assumption 1, the COLD/COLD-ORACLE algorithm (Case 1 or Case 2) with output $\mathbf{w}_t$, $t \in \mathbb{T}$ leads to the following average queue bound*

$$\frac{1}{T} \sum_{t=1}^{T} \bar{Q}[t-1] \quad \leq \quad \frac{V\delta'}{(\delta - \delta')} + \frac{1}{T(\delta - \delta')} \sum_{t=1}^{T} \Delta_t(\mathbf{w}_t, \mathbf{w}_{t,ref}) + \frac{2VD_{\Phi,\hat{\Phi}}[T]}{(\delta - \delta')}, \tag{13}$$

*where $\delta > \delta'$, $\bar{Q}[t-1] := \frac{1}{t-1} \sum_{k=1}^{t-1} Q_k[t-1]$, and $D_{\Phi,\hat{\Phi}}[T] = \frac{1}{T} \sum_{t=1}^{T} \sup_{\mathbf{w}} |\Phi_t(\mathbf{w}) - \hat{\Phi}_{t-1}(\mathbf{w})|$.*

First, note that the stability of the average queue length (boundedness) implies an asymptotic qualification of average constraint (see Neely & Huang (2010)). Hence, a bounded average queue size leads to good forgetting behavior. The bound in Theorem 2 admits a clear interpretation in terms of three distinct factors. The term $\frac{V\delta'}{\delta - \delta'}$ captures the *inherent difficulty of the CL problem*, reflecting task similarity and the tightness of the ideal feasibility tolerance. The cumulative deviation term $D_{\Phi,\hat{\Phi}}[T]$ quantifies *memory quality*, measuring how accurately the empirical loss approximates the true task losses across time. Finally, the drift term $\Delta_t(\cdot)$ represents *algorithmic smoothness*, encoding how aggressively the model parameters evolve between successive tasks. Together, these terms highlight the trade-off between problem hardness, memory fidelity, and update regularity in controlling long-term forgetting. The above result reveals that an average queue length scales as $\mathcal{O}(V)$ leading to a $\mathcal{O}(1/V)$ versus $\mathcal{O}(V)$ tradeoff between performance (Theorem 1) and the average queue size (Theorem 2). This is the standard trade-off arising in Lyapunov optimization / Drift-Plus-Penalty (DPP) frameworks, where (a) $\mathcal{O}(1/V)$ characterizes the optimality gap (performance), and (b) $\mathcal{O}(V)$ characterizes the queue size (constraint violation). This trade-off is well-established in the DPP literature, and our result shows that the proposed method inherits this canonical behavior in the CL setting. Further, the bound depends on $D_{\Phi,\hat{\Phi}}[T]$, which measures how the loss function obtained from the entire data set differs from the empirical loss obtained using the replay buffer in addition to the task variaton. In other words, larger memory size with similar data across tasks implies lower deviation, leading to a lower average queue length.

**Note:** An alternative metric commonly used in the CL literature evaluates performance via $\frac{1}{T} \sum_{t=1}^{T} \Phi_t(\mathbf{w}_T)$ and forgetting via $F_t := \frac{1}{t-1} \sum_{k=1}^{t-1} (\hat{\Phi}_k(\mathbf{w}_t) - \hat{\Phi}_k(\mathbf{w}_k))$ (see Lin et al. (2023)). The metric adapted in Lin et al. (2023) captures forgetting through accuracy degradation on past tasks; it is an endpoint-based measure. It does not capture how forgetting accumulates or is controlled during learning, which is exactly what our queue-based metric is designed to do. In particular, our queue-based metric captures the temporal evolution

of forgetting by penalizing sustained violations during learning. As a result, the two metrics are related but not equivalent. Further, our metric is more amenable to analysis with a particular emphasis on the tradeoff between stability and plasticity. We believe that our apporach provides further control and insights on the performance, complementing the existing work.

The exact minimization assumption in Theorems 1–2 follows the standard oracle model (for general non-convex losses) used in drift-plus-penalty analysis. These results characterize the intrinsic stability–optimality trade-off of the proposed formulation. In the next section, analyze gradient-based approximate optimization under smooth nonconvex assumptions, aligning the theory with practical deep learning settings.

## 6    Gradient-based Optimization for `COLD/COLD-ORACLE`

In the previous section, we provided theoretical guarantees on the performance of the `COLD-ORACLE` algorithm, demonstrating the trade-off between accuracy and constraint violation. In this section, we translate the DPP problem into an easily implementable iterative algorithm where a fixed learning rate and $V$ are used to achieve a good stationary point of the loss function in the current task and a lower forgetting. Towards solving (9), we use the following GD update on each task:

$$\mathbf{w}_{t+1} = \mathbf{w}_t - \eta \left( V \nabla \Phi_t(\mathbf{w}_t) + \sum_{k=1}^{t-1} Q_k[t-1] \nabla \hat{\Phi}_k(\mathbf{w}_t) \right), \tag{14}$$

where $\eta$ is a fixed learning rate, and the queue $Q_k[t-1]$ is obtained from the queue update in (5). In general, the convergence of GD updates depends on the learning rate through the smoothness constant of the function Orabona (2019). In our case, this corresponds to the smoothness of the $\text{DPP}_{V,t}(\mathbf{w})$ function, which is provided in the following lemma.

**Lemma 2.** *The function $\text{DPP}_{V,t}(\mathbf{w})$ is $\alpha_t := VL + \frac{l}{t-1} \sum_{k=1}^{t-1} Q_k[t-1]$-smooth.*

*Proof:* First, note that $\Phi_t(\mathbf{w})$ is $L$ smooth and $\Delta \hat{\Phi}_k(\mathbf{w}, \mathbf{w}_{t-1})$ is $l$ smooth. The proof follows from the fact that the sum of smooth functions is smooth, with a smoothness constant being the sum. $\square$

The current analysis assumes a fixed learning rate for simplicity, tractability and low algorithmic complexity. While this is standard in theoretical treatments, practical implementations often benefit from adaptive or scheduled learning rates. Extending the analysis to time-varying or adaptive learning rates is non-trivial, particularly due to their interaction with queue dynamics. We identify this as an important direction for future work. Next, we present the performance guarantees in terms of a bound on the average gradient when the domain $\mathcal{W}$ is compact with radius $r > 0$.[3]

**Theorem 3.** *Suppose Assumption 1 holds, and a fixed learning rate $\eta \leq 1/2\alpha_t$ for all $t$ is chosen. Then $\mathbf{w}_t$ obtained from (14) with $\tilde{\mathbf{w}}_{t,k}$ chosen according to Case 1 or Case 2 (i.e., `COLD/COLD-ORACLE`) results in*

$$\frac{1}{T} \sum_{t=1}^{T} ||\nabla \Phi_t(\mathbf{w}_t)||^2 \leq \frac{2\Phi_1(\mathbf{w}_1)}{TV\eta} + \frac{2D_\Phi[T]}{\eta V} + \frac{lr^2}{TV} \sum_{t=1}^{T} \rho_{t,-} + \frac{r^2 l}{\eta T V^2} \sum_{t=1}^{T} \bar{Q}[t-1], \tag{15}$$

*where $\rho_{t,-} := -\sum_{k=1}^{t-1} \rho_{t,k} \mathbf{1}\{\rho_{t,k} \leq 0\}$, $\rho_{t,k} := \inf_{\mathbf{w} \in \mathcal{W}} \langle \nabla \Phi_t(\mathbf{w}), \nabla \hat{\Phi}_k(\mathbf{w}) \rangle$, and $D_\Phi[t]$ is as given in Definition 1.*

The above result holds good for both reference models in Case 1 and Case 2. The last term corresponding to the queue average depends on the reference model used. The third term depends on the negative correlation, which measures the gradient correlation across tasks. In particular, $\rho_{t,k}$ captures the correlation of the gradients corresponding to tasks $t$ and $k$, and $\rho_{t,-}$ accumulates only the negative values of the correlation; higher $\rho_{t,-}$ means the tasks are highly negatively correlated. Further, the bound also reveals that using larger $\eta$ is beneficial. However, the choice of $\eta$ is limited by the constraint $\eta \leq 1/2\alpha_t$. Later, we provide more

---

[3]The compactness assumption is quite common in the literature on stochastic optimization and online learning (see Neely & Huang (2010)). Moreover, it is well known from the NTK analysis that SGD/GD updates of a deep neural network do not move out of a ball of a certain bounded radius Liu et al. (2022); Chizat & Bach (2019) making our results widely applicable.

insights on the effect of this on the overall performance. In the following, we provide a bound on the average queue for case 1 with step 6 of the `COLD-ORACLE` algorithm replaced by the GD update in (14). We use the following results to provide further insights into both forgetting and the average gradient performance of the GD update

**Theorem 4.** *The `COLD` (reference model in Case 1) algorithm with GD updates results in $\frac{1}{t-1}\sum_{k=1}^{t-1} Q_k[t-1] = \mathcal{O}(V)$, and the following average queue bound*

$$\frac{1}{T}\sum_{t=1}^{T}\bar{Q}[t-1] \;\leq\; \frac{VD_\Phi[T]}{\delta} + \frac{1}{T\delta}\sum_{t=1}^{T}\Delta_t(\mathbf{w}_t, \mathbf{w}_{t-1}), \tag{16}$$

*where $\bar{Q}[t-1]$ is as defined in Theorem 2.*

Substituting the bound on the queue in Theorem 4 in the result of Theorem 3, we get the following corollary.

**Corollary 1.** *Suppose Assumption 1 holds with compact $\mathcal{W}$, and a fixed learning rate $\eta \leq 1/2\alpha_t$ for all $t$ is chosen. Then for $\mathbf{w}_t$ updates according to (14) with $\tilde{\mathbf{w}}_{t,k} = \mathbf{w}_{t-1}$ as in case 1 (`COLD`), we have*

$$\frac{1}{T}\sum_{t=1}^{T}||\nabla\Phi_t(\mathbf{w}_t)||^2 \leq \frac{2\Phi_1(\mathbf{w}_1)}{TV\eta} + \frac{D_\Phi[T]}{\eta V}\left(2 + \frac{r^2 l}{V\delta}\right) + \frac{lr^2}{TV}\sum_{t=1}^{T}\rho_{t,-} + \frac{r^2 l}{\delta\eta TV^2}\sum_{t=1}^{T}\Delta_t(\mathbf{w}_t, \mathbf{w}_{t-1}),$$

*where $\rho_{t,-} := -\sum_{k=1}^{t-1}\rho_{t,k}\mathbf{1}\{\rho_{t,k} \leq 0\}$.*

In the above corollary, the term $\rho_{t,-}$ quantifies intrinsic task interference through negative gradient alignment between past and current tasks, indicating regimes in which forgetting is unavoidable. This gradient correlation is a fundamental quantity in CL, as it captures how learning on the current task perturbs representations learned for previous tasks. As mentioned in 1, projection-based methods such as GEM and `A-GEM` explicitly exploit this correlation by constraining updates to lie in the subspace where gradients of past and current tasks are non-negatively aligned, leading to mitigating the instantaneous interference. However, while correcting locally conflicting updates only leads to short-term improvements, our analysis highlights that $\rho_{t,-}$ is a sustained negative alignment across tasks leads to cumulative forgetting, motivating the proposed queue-based control formulations that regulate interference over time rather than per step. Our analysis demonstrates that sustained negative alignment increases the virtual queue and is therefore penalized over time, ensuring that task interference captured by $\rho_{t,-}$ is accounted for at the level of long-term stability rather than instantaneous updates. It is also important to note that for a a fixed $\eta$, there is an inherent $V$ and $1/V$ tradeoff between queue stability and performance, which is not an artifact of the optimization approach. The above corollary requires $\eta \leq 1/2\alpha_t$ for all $t$. Towards ensuring this, we use the following bound on $\alpha_t$ for our choice of $\eta$:

$$\alpha_t = VL + \frac{l}{t-1}\sum_{k=1}^{t-1} Q_k[t-1] \overset{(a)}{\leq} VL + \mathcal{O}(V) = \mathcal{O}(V) \tag{17}$$

where $(a)$ follows from the first queue bound in Theorem 4. Using $\eta = \mathcal{O}\left(1/V\right)$ in the above corollary, and the queue bound for case 1, we get the following trade-off.

**Corollary 2.** *Suppose Assumption 1 holds with compact $\mathcal{W}$, and a fixed learning rate $\eta = \mathcal{O}\left(\frac{1}{V}\right)$ is chosen. Then for $\mathbf{w}_t$ updates according to (14) with $\tilde{\mathbf{w}}_{t,k} = \mathbf{w}_{t-1}$ as in case 1 (`COLD`), we have*

$$\frac{1}{T}\sum_{t=1}^{T}||\nabla\Phi_t(\mathbf{w}_t)||^2 \leq \mathcal{O}\left(\frac{1}{T} + D_\Phi[T] + \frac{\bar{\rho}}{V} + \frac{\bar{\Delta}}{\delta V}\right), \tag{18}$$

*where $\bar{\rho} := \frac{1}{T}\sum_{t=1}^{T}\rho_{t,-}$, $\bar{\Delta} := \frac{1}{T}\sum_{t=1}^{T}\delta_t(\mathbf{w}_t, \mathbf{w}_{t-1})$, and $\delta > \delta'$. The corresponding queue bound is*

$$\frac{1}{T}\sum_{t=1}^{T}\bar{Q}[t-1] \leq \mathcal{O}\left(\frac{VD_\Phi[T]}{\delta}\right). \tag{19}$$

Even with one step GD, the above corollary reveals the $V$ versus $1/V$ tradeoff between the average gradient squared and the average queue. The performance of the method depends on the choice of $V$, which governs the plasticity–stability trade-off. While our analysis provides theoretical guidance, designing mechanisms to adapt $V$ automatically during training remains an open problem, and we identify it as an important direction for future work. Further, when the term $D_\Phi[T]$ is negligible, then larger number of tasks $T$ ensures a smaller gradient average while keeping the queue bounded. The proposed algorithm employs a single step of GD, which results in task variation dominating both the performance guarantee and the average queue bound. We conjecture that performing multiple GD steps per task could mitigate the impact of variation; investigating this extension is left for future work. Next, we show that bounded average queue above implies asymptotic satisfaction of the constraints.

**Bounded average queue:** Suppose $\frac{1}{T}\sum_{t=1}^{T}\bar{Q}[t-1] < \infty$ for all $T$, then the term $\bar{Q}[t-1] < \infty$ for infinitely many $t$. Since $\bar{Q}[t-1] = \frac{1}{t-1}\sum_{k=1}^{t-1}Q_k[t-1]$, it follows that $Q_k[t-1] < \infty$ for infinitely many $t$. Note that $Q_k[t-1] \geq Q_k[t-2] + \hat{\Phi}_k(\mathbf{w}_t) - \hat{\Phi}_k(\mathbf{w}_{t-1})$, which implies that $\frac{1}{t-1}\sum_{k=1}^{t-1}(\hat{\Phi}_k(\mathbf{w}_t) - \hat{\Phi}_k(\mathbf{w}_{t-1}) - \delta) \leq \frac{Q_k[t-1]}{t-1} = \mathcal{O}\left(\frac{1}{t-1}\right)$ for infinitely many $t$. Thus, for larger values of $t$, the average constraint is satisfied approximately (except for the $1/(t-1)$ factor) for infinitely many $t$.

**Space/computation time complexity and Limitations:** `COLD` maintains a computational complexity of $O((b+m)\cdot C)$, comparable to standard replay methods, while introducing only a negligible overhead of $O(t)$ scalar storage for virtual queues. Here, $b$ is the current task batch size, $m$ is the memory batch size per task and $C$ is the per-sample gradient computation cost. The full `COLD-ORACLE` variant offers enhanced constraint expressivity by incorporating task-wise replay, at the cost of $O((b+t\cdot m)\cdot C)$ computation and $O(t\cdot d)$ additional storage, where $d$ the dimension. Crucially, both variants replace expensive projection (see Lopez-Paz & Ranzato (2017)) or meta-learning steps (see Riemer et al. (2019)) with a lightweight queue-based reweighting mechanism, enabling explicit control of forgetting with minimal computational overhead. A more detailed analysis can be found in the appendix. In addition, we have used fixed learning rate and constant $V$. It is beneficial to use varying learning rate and $V$ adapted to various parameters of the systems. Designing an adaptive learning rate and $V$ is a limitation of this work, and will be studied as a part of the future work. Further, using multiple rounds of GD may help achieve better trade-off between plasticity and stability. However, to keep the algorithm simple and tractable, such studies are not done here, and is relegated to the future work.

## 7 Experimental Results and Discussions

In this section, we present the experimental results to corroborate our theory and demonstrate the efficacy of the proposed `COLD` (*Case* 1) and `COLD-ORACLE` (*Case* 2) algorithms on standard benchmark datasets in the task-incremental learning setup.

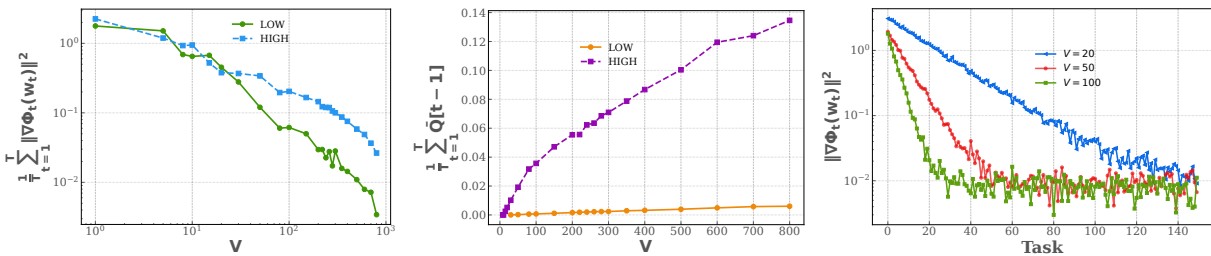

Figure 1: Toy quadratic CL setup (`COLD` algorithm) with controlled task generation. **Left:** Average gradient squared versus $V$ in log-log scale. The variation is linear. **Center:** Average queue size versus $V$ in the linear scale. **Right:** Gradient squared versus task $t$. Results are shown for both LOW (near-IID) and HIGH (non-IID with drift) task variation regimes. The left figure highlights the effect on plasticity while center figure shows the effect on forgetting.

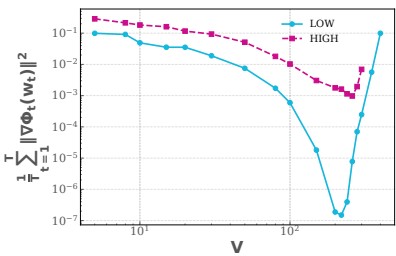

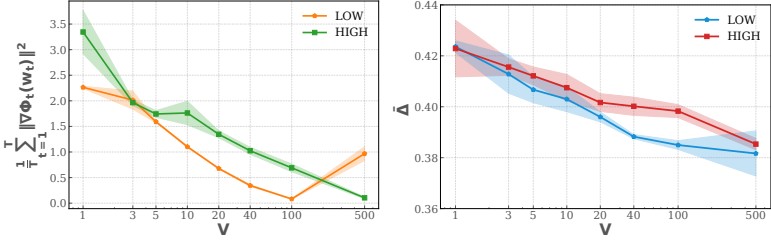

Figure 2: Average gradient squared versus $V$ in the log-log scale for high and low task variations demonstrating the effect of $\eta$.

Figure 3: Average norm square gradient and $\bar{\Delta}$ versus $V$ in the high and low task variation regimes for `COLD` on Split-MNIST data set.

## 7.1 Theoretical Validation

To validate the theoretical result in Corollary 1, we analyze `COLD` for a controlled toy model with a quadratic loss under two regimes of task variation (low and high). In this setup, each task corresponds to minimizing a quadratic objective centered at a task-specific optimum $\mathbf{w}_t^*$, i.e., learning reduces to tracking a sequence of shifting optima. The base task is defined by a reference optimum $\mathbf{w}_{\text{base}}^*$, and subsequent tasks are generated as perturbations or structured drifts from this base optimum. Thus, task variation is entirely governed by how $\mathbf{w}_t^*$ evolves across tasks starting from the base task. In particular, we consider $T$ CL tasks in $\mathbb{R}^d$, where each task $t$ is defined by the quadratic objective $\Phi_t(\mathbf{w}) = \frac{1}{2}\|\mathbf{w} - \mathbf{w}_t^*\|^2$. In the following, we explain the two regimes in detail.

**Low Task Variation (Near-IID, LOW):** In this regime, tasks share a common optimum with small perturbations: $\mathbf{w}_t^* = \mathbf{w}_{\text{base}}^* + \boldsymbol{\varepsilon}_t$, where $\mathbf{w}_{\text{base}}^* \sim \mathcal{N}(\mathbf{0}, c_1\mathbf{I}_d)$ and $\boldsymbol{\varepsilon}_t \sim \mathcal{N}(\mathbf{0}, c_2\mathbf{I}_d)$. Here, we choose $c_1 = 0.09$ and $c_2 = 0.0004$, which yields $D_\Phi[T] \approx 0.02$.[4]

**High Task Variation (Non-IID with Drift, HIGH):** In this regime, the tasks drift linearly along a random direction with additional noise. More precisely, $\mathbf{w}_t^* = \mathbf{w}_{\text{base}}^* + \frac{t}{T} \cdot 0.8 \cdot \mathbf{v} + \boldsymbol{\xi}_t$, where $\mathbf{w}_{\text{base}}^* \sim \mathcal{N}(\mathbf{0}, c_1\mathbf{I}_d)$, $\boldsymbol{\xi}_t \sim \mathcal{N}(\mathbf{0}, c_2\mathbf{I}_d)$ and $\mathbf{v} \sim \mathcal{N}(\mathbf{0}, \mathbf{I}_d)$ is normalized to get $\mathbf{v} \leftarrow \mathbf{v}/\|\mathbf{v}\|$. Here, we choose, $c_1 = 0.09$ and $c_2 = 0.0064$, which yields $D_\Phi[T] \approx 0.08$. Both regimes use $d = 25$ and $T = 150$ tasks.

The task variation $D_\Phi[T]$ is measured empirically using Definition 1. The above settings imply $\hat{\Phi}_t(\mathbf{w}) = \Phi_t(\mathbf{w})$ for all $t \in [T]$ as the above toy example does not require replay buffer. To validate the proposed theoretical results, we have fixed $\eta = 4 \times 10^{-4}$, and $\delta = 4 \times 10^{-7}$, and ran the `COLD-ORACLE` algorithm with 3 GD steps per task.[5] The Fig. 1 (**Left**) shows a plot of the average gradient square versus $V$ in the log-scale. From Corollary 1, as $V$ increases, the average gradient squared scales down as $1/V$, i.e., it decreases linearly in the log scale, as depicted in the figure. Further, the figure also demonstrates that higher variation results in a lower slope, as predicted by our theory. The second figure (**Center**) shows the plot of the average queue size versus $V$ in the linear scale. We observe that the average queue size increases linearly with $V$, and the slope increases with the variation, this is inline with our theoretical prediction. The third figure (**Right**) shows a plot of the gradient squared versus $t$. For larger $t$, the gradient is saturating due to the residual variation term. In particular, larger $V$ is saturating fast while $V = 50$ requiring a greater number of tasks to saturate, as suggested by our result. The first two figures illustrate the canonical $\mathcal{O}(1/V)$–$\mathcal{O}(V)$ tradeoff, making explicit that no single choice of $V$ is uniformly optimal: larger values favor stability at the expense of plasticity, while smaller values induce the opposite behavior. Crucially, our objective is not to design or optimize $V$. In the DPP framework, $V$ is an externally specified control parameter that encodes the desired stability–plasticity operating point dictated by the application, as in the classical DPP framework Neely et al. (2012). Once this requirement is fixed, an appropriate $V$ can be selected using standard empirical tuning procedures, which is orthogonal to the theoretical contributions of this work. Our theory assumes $\eta \leq \frac{1}{2\alpha_t} = \mathcal{O}\left(\frac{1}{V}\right)$. For a fixed $\eta$, increasing $V$ beyond a threshold violates the condition, resulting in an increased gradient. This is demonstrated in Fig. 2, where the average gradient squared increases beyond

---

[4]A closed-form expression for $D_\Phi[T]$ can easily be obtained in the case of quadratic loss.

[5]This can be reduced to 1 without changing the trend.

Table 1: Performance comparison on benchmark datasets under different memory buffer sizes ($M$). Standard deviation values are provided in the appendix. '–' denotes not applicable, as forgetting is not defined for GDumb due to its training strategy Prabhu et al. (2020).

| Method | Split-CIFAR10 | | | | Split-CIFAR100 | | | | Split-TinyImageNet | | | |
|---|---|---|---|---|---|---|---|---|---|---|---|---|
| | $M = 0.6$k | | $M = 1$k | | $M = 1$k | | $M = 5$k | | $M = 2$k | | $M = 5$k | |
| | ACC(%) | F | ACC(%) | F | ACC(%) | F | ACC(%) | F | ACC(%) | F | ACC(%) | F |
| Fine-tune | 76.70 | 0.22 | 76.70 | 0.22 | 29.86 | 0.54 | 29.86 | 0.54 | 16.59 | 0.44 | 16.59 | 0.44 |
| EWC(2017) | 71.05 | 0.35 | 71.05 | 0.35 | 35.00 | 0.17 | 35.00 | 0.17 | 24.12 | 0.12 | 24.12 | 0.12 |
| A-GEM(2018) | 83.75 | 0.42 | 83.81 | 0.43 | 63.59 | 0.30 | 69.82 | 0.33 | 51.11 | 0.23 | 53.43 | 0.25 |
| MER(2019) | 77.32 | 0.04 | 76.91 | 0.02 | 47.35 | 0.05 | 49.68 | 0.05 | 34.89 | 0.05 | 53.76 | 0.01 |
| GDumb(2020) | 64.68 | – | 63.01 | – | 37.56 | – | 51.42 | – | 21.00 | – | 57.34 | – |
| ER(2019) | 79.31 | 0.09 | 81.98 | 0.02 | 57.60 | 0.08 | 68.56 | 0.34 | 23.89 | 0.03 | 49.34 | 0.15 |
| DER(2020) | 83.66 | 0.05 | 85.05 | 0.04 | 50.27 | 0.22 | 48.31 | 0.24 | 28.22 | 0.20 | 30.58 | 0.18 |
| DER++(2020) | 84.84 | 0.03 | 85.80 | 0.03 | 66.96 | 0.07 | 71.67 | 0.04 | 56.19 | 0.05 | 58.88 | 0.04 |
| ER-ACE(2022) | 82.70 | 0.03 | 86.11 | 0.02 | 64.97 | 0.05 | 70.14 | 0.04 | 51.37 | 0.07 | 55.67 | 0.05 |
| CBA(2023) | 85.01 | 0.02 | 84.53 | 0.01 | 63.70 | 0.03 | 73.75 | 0.02 | 55.92 | 0.04 | 58.92 | 0.03 |
| NCCL(2023) | 72.99 | 0.25 | 71.90 | 0.24 | 29.20 | 0.54 | 50.79 | 0.35 | 19.64 | 0.44 | 30.15 | 0.33 |
| REFRESH(2024) | 84.93 | 0.03 | 85.87 | 0.03 | 67.34 | 0.06 | 70.84 | 0.04 | 56.68 | 0.06 | 60.08 | 0.03 |
| COLD (Ours) | 84.90 | 0.07 | 85.90 | 0.09 | **68.03** | 0.10 | 74.56 | 0.05 | 55.09 | 0.10 | 61.39 | 0.07 |
| COLD-ORACLE (Ours) | **85.68** | 0.06 | **87.94** | 0.05 | 67.38 | 0.18 | **75.64** | 0.05 | **56.74** | 0.08 | 62.21 | 0.02 |

$V = 200$. Finally, we demonstrate in Fig. 3 that large variations (non-iid) decrease the rate of convergence of the average gradient squared compared to small variations (iid) even in the practical setting with non-convex loss using Split-MNIST Deng (2012) data set. In particular, we depict the average gradient squared for the near-IID (LOW) and non-IID (HIGH) settings, along with $\bar{\Delta}$, which is the residual term in the convergence bound measuring the average error incurred due to the algorithm updating the model before fully aligning with the current task optimum. We observe that both quantities, the average gradient squared and the average algorithmic error, exhibit consistent trends: under high task variation, both remain elevated, whereas in the near-IID regime they decrease and stabilize at lower values.

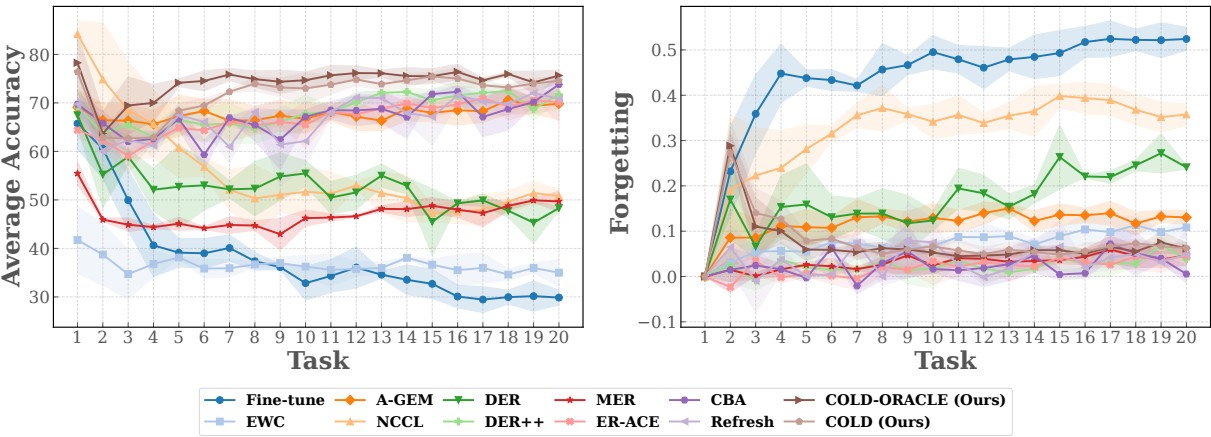

Figure 4: Baseline comparison: average accuracy and forgetting versus tasks on Split-CIFAR100.

## 7.2 Empirical Evaluations

In this subsection, we carry out extensive empirical evaluation and present comparisons with relevant baselines and ablation studies on well-known CL benchmarks. Specifically, we examine the following aspects in practical settings: (a) How well does the proposed technique perform as compared to the baselines? (b) Trade-off in $V$, (c) Stability of the queue, and (d) factors that affect plasticity. [6]

---

[6]The code is available at https://github.com/nazreenshah/COLD.

### 7.2.1 Experimental Setup

**Datasets:** We demonstrate the experimental results on CL benchmarks such as Split-CIFAR10, Split-CIFAR100 Krizhevsky & Hinton (2009), and Split-TinyImageNet Le & Yang (2015) to evaluate the task-incremental setting with disjoint class splits and known task identities. Details of the datasets are provided in the appendix. We also evaluate on Permuted-MNIST (PMNIST) to study the domain-incremental setting, where tasks share the same label space while the input distributions differ across tasks; results are reported in the appendix.

**Architecture:** We use the following architectures depending on the dataset: (a) Split-MNIST: an MLP with a single hidden layer of 256 units and a task-specific multi-head output; (b) Split-CIFAR10, Split-CIFAR100, and Split-TinyImageNet: a standard ResNet-18 with a task-specific multi-head classifier; and (c) PMNIST: a two-layer fully connected network with 256 units in each layer using a single shared output head across tasks.

**Metrics**: We assess the performance of the proposed algorithms using the model's test accuracy on a sequence of $T$ tasks, and the forgetting, which are defined below: (a) **Average Accuracy** (ACC $\in [0,1]$): The average accuracy is computed using ACC $:= \frac{1}{T}\sum_{j=1}^{T} a_{T,j}$, where $a_{t,j}$ is the accuracy of the model obtained after observing task $t$ on task $j < t$. (b) **Forgetting:** We capture the forgetting factor using $F := \frac{1}{T-1}\sum_{j=1}^{T-1}\left[\max_{l\in[T-1]}(a_{l,j} - a_{T,j})\right]$, where the difference $\max_{l\in[T-1]}(a_{l,j} - a_{T,j})$ captures the extent to which previously learned knowledge is overwritten by subsequent learning. Experiments are carried out on 3 multiple runs with a fixed set of random seeds. We report the mean and standard deviation ($\pm$) across these runs in the appendix.

### 7.2.2 Comparison with Baselines

We evaluate the proposed `COLD-ORACLE` and `COLD` methods against a broad and representative set of CL baselines encompassing regularization-based, replay-based, and optimization-based approaches. Specifically, we compare against: (a) `Fine-tune`; (b) `EWC` Kirkpatrick et al. (2017); (c) `A-GEM` Chaudhry et al. (2018); (d) `NCCL` Han et al. (2023); (e) `MER` Riemer et al. (2019); (f) `GDumb` Prabhu et al. (2020); (g) `DER` and `DER++` Buzzega et al. (2020); (i) `ER-ACE` Caccia et al. (2021); (j) `CBA` Wang et al. (2023); and (k) `REFRESH` Wang et al. (2024). Detailed descriptions of all baselines are provided in the appendix.

From Table 1, we observe that `COLD-ORACLE` consistently achieves the highest average accuracy across all benchmark datasets and memory budgets, while `COLD` attains the second-best accuracy in most settings. This demonstrates that the proposed framework is particularly effective at preserving task-relevant knowledge as the number of tasks grows. Importantly, while the forgetting values of `COLD-ORACLE` and `COLD` are sometimes marginally higher than those of strongly stability-biased methods, they remain well-controlled and competitive across all benchmarks. This behavior is expected, since the proposed methods place greater emphasis on plasticity and forward transfer, leading to improved final-task performance and overall accuracy at the cost of increased forgetting. In contrast, several baselines achieve lower forgetting primarily by constraining updates aggressively, which often leads to substantially degraded accuracy. Our results highlight that minimizing forgetting alone is insufficient; maintaining high predictive performance across tasks is equally critical.

Consistent trends are also observed in Fig. 4, where the proposed methods outperform competing approaches as learning progresses. Regularization-based methods such as `EWC` exhibit limited forgetting but suffer from poor accuracy, while optimization-based methods such as `NCCL` degrade rapidly as the number of tasks increases. Replay-based approaches, including recent plug-in methods such as `CBA` and `REFRESH`, improve performance but remain consistently inferior to `COLD-ORACLE`, despite their reliance on enhanced replay mechanisms.

Overall, these results demonstrate that `COLD-ORACLE` and `COLD` achieve a more favorable accuracy–forgetting tradeoff, delivering state-of-the-art accuracy with stable and controlled forgetting, particularly in long task sequences where plasticity is essential.

### 7.2.3 Ablation and Sensitivity Analysis

We next examine the impact of varying the memory buffer size $|\mathcal{M}|$, the number of epochs per task, the number of tasks, the parameters $\delta$ and $V$ on the resulting accuracy-forgetting trade-offs and queue stability.

Each experiment in the following section is presented on a representative dataset, while corresponding results on additional datasets are deferred to the appendix.

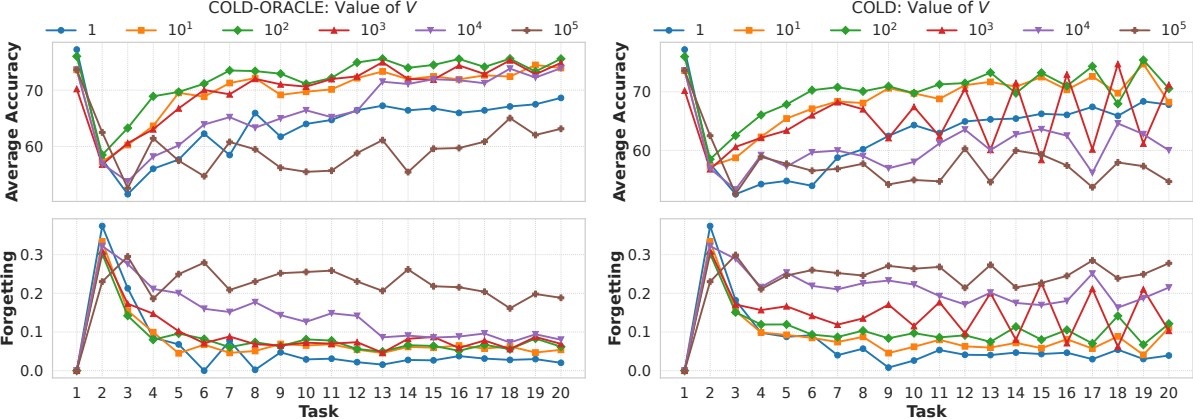

Figure 5: Trade-off between average accuracy and forgetting with varying $V$ on Split-CIFAR100 using `COLD-ORACLE` (left) and `COLD` (right).

**Accuracy versus Forgetting - Varying $V$:** Recall that $V$ balances the trade-off between average accuracy and forgetting. In Fig. 5, we study this trade-off by varying $V$ from 1 to $10^5$ on both the proposed algorithms. Evidently $V = 10^2$ provides better average accuracy but lower forgetting for the Split-CIFAR100 dataset. In contrast, $V = 1$ leads to the least forgetting, but results in poor average accuracy. Hence, $V$ can be used as a tuning parameter to achieve the desired requirements in CL. It can also be noted that `COLD-ORACLE` benefits from a balanced stability–plasticity trade-off and demonstrates stable results across tasks even when $V$ is large.

**Varying memory buffer size ($M$):** We analyze the effect of varying the total memory size $M$ by varying memory samples per task in the proposed algorithms, ranging from 50 to 500 samples per task. We observe from Fig. 6 (**Left**) that for Split-CIFAR100, as the sampling size increases, there is an improvement in average accuracy (line plot) and a decrease in forgetting (bar plot). We see a similar trend for both `COLD-ORACLE` and `COLD` algorithms across all datasets (see appendix). Notably, the performance gap between `COLD-ORACLE` and `COLD` reduces as $M$ increases. This is because `COLD` relies on replay-based gradients, which introduce estimation noise that scales as $O(1/M)$. For small $M$, this noise leads to weaker constraints compared to `COLD-ORACLE`'s best-model reference. However, as $M$ grows, the replay estimate becomes increasingly accurate, making the constraints imposed by both methods nearly identical, thereby diminishing the gap.

**Varying $\delta$-parameter:** Fig. 6 (**Right**) examines the effect of the hyperparameter $\delta$, which controls the tightness of the constraint in the optimization problem in (3). In principle, $\delta = 0$ enforces a strict constraint, while larger values of $\delta$ correspond to progressively relaxed constraints. Empirically, however, we observe that larger values (e.g., $\delta = 10$) result in degraded performance in terms of both average accuracy and forgetting for `COLD-ORACLE` and `COLD`, compared to smaller $\delta$. At first glance, this may appear counter-intuitive. This behavior can be attributed to the fact that our theoretical guarantees are derived with respect to the queue-based notions of accuracy and forgetting, whereas the experimental metrics employ a different operational definition. Consequently, excessive relaxation of the constraint weakens the effective control exerted by the DPP mechanism under the empirical metrics, leading to poorer observed performance. Additionally, we observe that `COLD` exhibits higher instability across tasks compared to `COLD-ORACLE`. This stems from its use of the previous model as the reference, which changes at every task and causes the queue updates to depend on a moving target. As a result, errors can propagate across tasks, leading to fluctuations in performance. In contrast, `COLD-ORACLE` uses the best historical model as a stable reference, which mitigates such error accumulation and results in smoother behavior.

**Impact of $T$:** In this experiment, we investigate how well the CL method scales with the number of tasks $T$-this helps us to understand the model's ability to retain earlier knowledge while learning new tasks. From

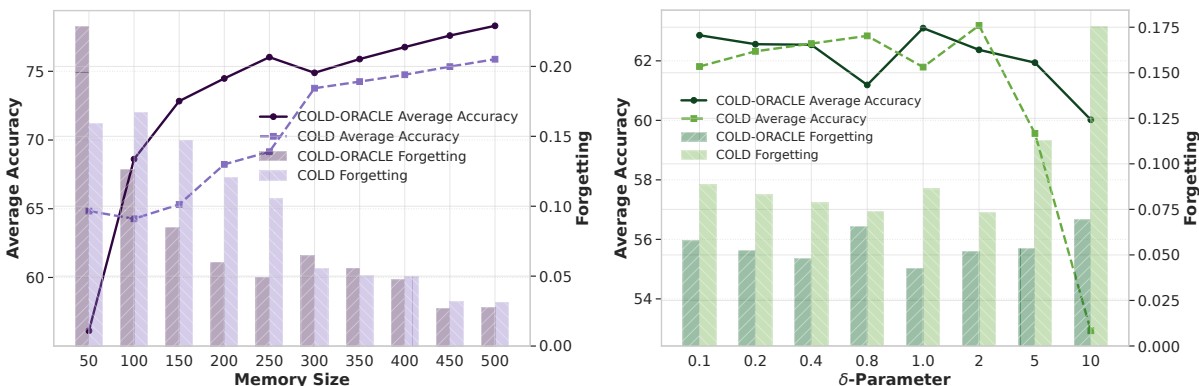

Figure 6: **Left:** Effect of memory budget $M$. **Right:** Effect of varying $\delta$ parameter using `COLD-ORACLE` and `COLD`.

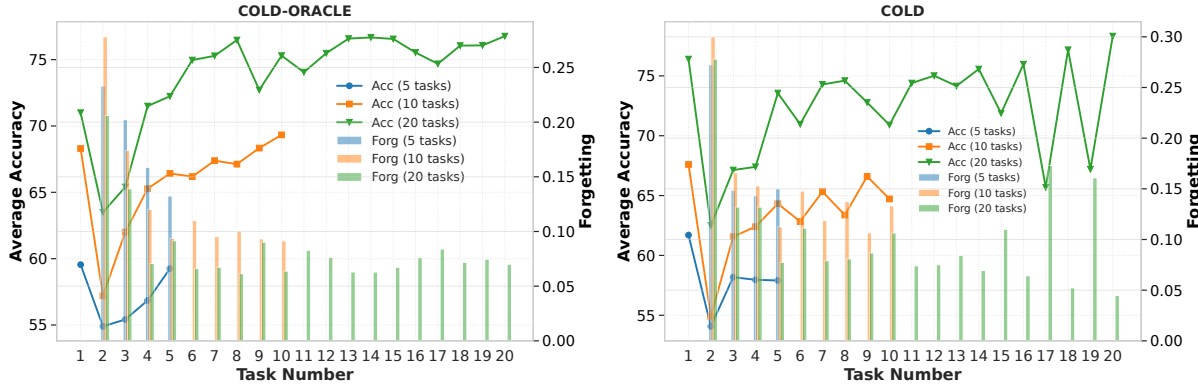

Figure 7: Scalability with increasing number of tasks on Split-CIFAR100 using `COLD-ORACLE` (left) and `COLD` (right).

Fig. 7, we observe that `COLD-ORACLE` (**Left**) remains stable with respect to both accuracy and forgetting even as the number of tasks increases. We see a similar trend using the `COLD` (**Right**) algorithm, but with a less stable performance across tasks.

**Queue Stability Analysis:** In Fig. 8, we validate the empirical queue stability of `COLD-ORACLE` and `COLD` by observing that a smaller $V = 1$ leads to smaller queue values, and vice versa for $V = 10^2$ and $V = 10^4$. Among different values of $V$, using $V = 10^2$ seems to balance the model updates on both current and past data as the color is neither too dark as in $V = 10^4$ nor too light as in $V = 1$. Note that $V = 10^4$ places greater weight on the current data, destabilizing the queue and leading to greater catastrophic forgetting, as also observed in Fig. 5. While both methods exhibit similar qualitative trends with respect to $V$, `COLD` consistently maintains lower queue values than `COLD-ORACLE`. This behavior can be attributed to the choice of reference model: `COLD` uses the immediately preceding model as the reference, leading to incremental constraint enforcement, whereas `COLD-ORACLE` relies on the best available model, which imposes stricter constraints and results in larger accumulated queue values.

## 8 Conclusions and Future Work

In this work, we introduced `COLD`, a Lyapunov optimization–based framework for continual learning that provides explicit and provable guarantees on the trade-off between current task accuracy and catastrophic forgetting. To support analysis and benchmarking, we additionally considered `COLD-ORACLE`, an oracle reference model representing an idealized setting. By formulating CL as a long-term constrained optimization

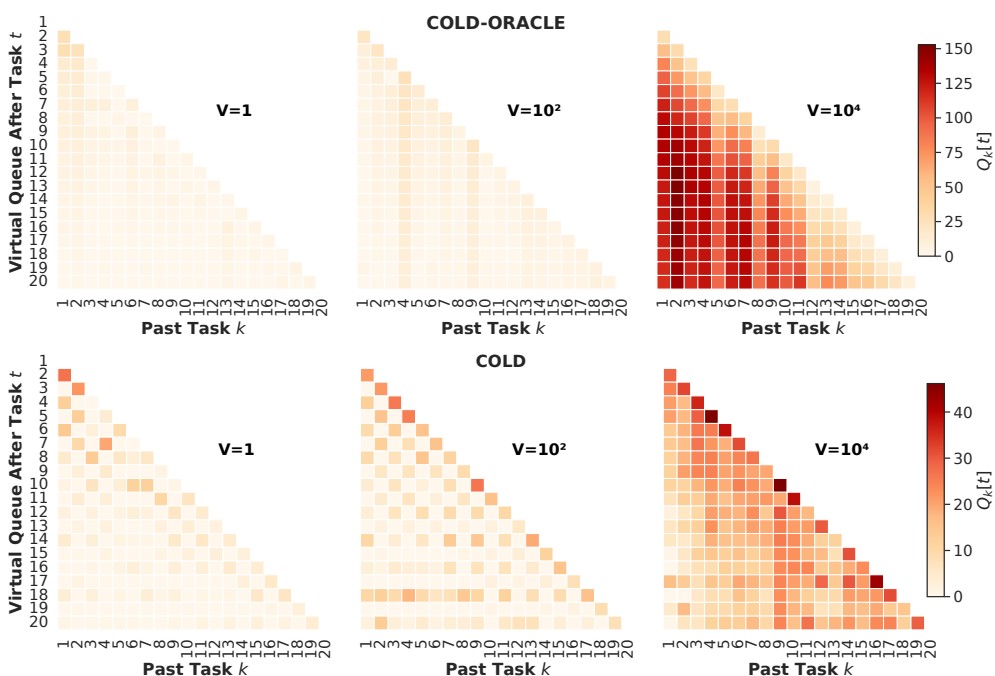

Figure 8: Virtual Queue for varying $V$ on Split-CIFAR100 using **Top:** `COLD-ORACLE` and **Bottom:** `COLD`.

problem, we showed that performance on the current task and forgetting of past tasks can be jointly controlled through a single tunable parameter, yielding interpretable bounds that hold at every task rather than only asymptotically or in hindsight.

A key outcome of our analysis is the explicit dependence of the regret and forgetting guarantees on task variation, thereby revealing how non-stationarity across tasks fundamentally impacts CL performance. This connection is notably absent in existing theoretical work, which typically treats tasks as exchangeable or ignores their temporal evolution altogether. Our results demonstrate that no algorithm can simultaneously achieve low regret and negligible forgetting when task variation is large, and that our method adapts optimally to this regime.

From an algorithmic perspective, both `COLD` and `COLD-ORACLE` enforce long-term constraints via virtual queues, eliminating the need for projection steps or dual-variable updates commonly used in constrained CL methods. The proposed algorithm admits fixed learning rates and remains simple to implement, while still enjoying rigorous performance and stability guarantees.

Overall, this work establishes a principled control-theoretic foundation for CL, bridging online optimization, queue stability, and learning under nonstationarity. We believe this framework opens several promising directions for future research, including tighter variation-dependent bounds, extensions to nonconvex objectives, usage of adaptive learning rates and $V$, and practical implementations in large-scale and task-agnostic CL settings.

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

## A    Theoretical Appendix

Here we provide the theoretical analysis underpinning the proposed framework, including additional Lemmas, proofs of the main results, and detailed derivations omitted from the main paper.

**Lemma 3.** *Assume that the domain of the loss functions $\mathcal{W}$ is compact with radius $r > 0$. The optimality gap $\hat{\Phi}_k(\mathbf{w}) - \hat{\Phi}_k(\mathbf{v}_k^*)$ is bounded by $\frac{r^2 l}{2}$ and the optimality gap $\Phi_k(\mathbf{w}) - \Phi_k(\mathbf{u}_k^*)$ is bounded by $\frac{r^2 L}{2}$ for all $\mathbf{w} \in \mathcal{W}$.*

*Proof:* It follows from the smoothness of $\hat{\Phi}_k(\mathbf{w})$, the fact that $\nabla \hat{\Phi}_k(\mathbf{v}^*) = 0$, and the fact that $||\mathbf{w}_t - \mathbf{u}_k^*||^2 \leq r^2$ (due to the compactness assumption). The second inequality can be proved in a similar manner.

### A.1    Proof of Lemma 1

From the definition of queue updates, and using $\max[x, 0]^2 \leq x^2$ gives

$$Q_k[t]^2 \leq Q_k[t-1]^2 + (\Delta\hat{\Phi}_k(\mathbf{w}, \tilde{\mathbf{w}}_{t,k}))^2 + 2Q_k[t-1]\Delta\hat{\Phi}_k(\mathbf{w}, \tilde{\mathbf{w}}_{t,k}). \tag{20}$$

Dividing by $2(t-1)$ and summing from $k = 1$ to $t - 1$ results in

$$\Delta\mathcal{L}[t] \quad \leq \quad \Delta_t(\mathbf{w}, \mathbf{w}_{t,\mathrm{ref}}) + \frac{1}{t-1}\sum_{k=1}^{t-1} Q_k[t-1]\Delta\hat{\Phi}_k(\mathbf{w}, \tilde{\mathbf{w}}_{t,k}),$$

where $\Delta_t(\mathbf{w}, \mathbf{w}_{t,\mathrm{ref}}) := \frac{1}{2(t-1)}\sum_{k=1}^{t-1}(\Delta\hat{\Phi}_k(\mathbf{w}, \tilde{\mathbf{w}}_{t,k}))^2$, and $\mathbf{w}_{t,\mathrm{ref}}$ is as defined in the Lemma. □

## B    Proof of Theorem 1

Using the drift bound established in Lemma 1, we can upper bound the DPP term at task $t$ as follows:

$$V\Phi_t(\mathbf{w}_t) + \Delta\mathcal{L}[t] \quad \leq \quad V\Phi_t(\mathbf{w}_t) + \Delta_t(\mathbf{w}_t, \mathbf{w}_{t,\mathrm{ref}}) + \frac{1}{t-1}\sum_{k=1}^{t-1} Q_k[t-1]\Delta\hat{\Phi}_k(\mathbf{w}_t, \tilde{\mathbf{w}}_{t,k})$$

$$\leq \quad V\Phi_t(\mathbf{w}_t^*) + \Delta_t(\mathbf{w}_t, \mathbf{w}_{t,\mathrm{ref}}) + \frac{1}{t-1}\sum_{k=1}^{t-1} Q_k[t-1]\Delta\hat{\Phi}_k(\mathbf{w}_t^*, \tilde{\mathbf{w}}_{t,k}), \tag{21}$$

where $\Delta_t(\mathbf{w}_t, \mathbf{w}_{t,\mathrm{ref}}) := \frac{1}{2(t-1)}\sum_{k=1}^{t-1}(\hat{\Phi}_k(\mathbf{w}_t) - \hat{\Phi}_k(\tilde{\mathbf{w}}_{t,k}) - \delta)^2$, and the second inequality follows since $\mathbf{w}_t \in \arg\inf_{\mathbf{w}} V\Phi_t(\mathbf{w}) + \sum_{k=1}^{t-1} Q_k[t-1]\Delta\hat{\Phi}_k(\mathbf{w}, \tilde{\mathbf{w}}_{t,k})$ is an optimal solution for the DPP, and hence using $\mathbf{w}_t^*$ results in an upper bound. Recall that $\Delta\hat{\Phi}_k(\mathbf{w}_t^*, \tilde{\mathbf{w}}_{t,k}) := \hat{\Phi}_k(\mathbf{w}_t^*) - \hat{\Phi}_k(\tilde{\mathbf{w}}_{t,k}) - \delta \leq \hat{\Phi}_k(\mathbf{w}_t^*) - \inf_{\mathbf{w}} \hat{\Phi}_k(\mathbf{w}) - \delta \leq \delta' - \delta$, $k = 1, 2, \ldots, t-1$, where $\mathbf{w}_t^*$ is the optimal solution to (9). Moreover, $\delta > \delta'$, and hence the last term becomes negative, which results in the following bound

$$V\Phi_t(\mathbf{w}_t) + \Delta\mathcal{L}[t] \leq V\Phi_t(\mathbf{w}_t^*) + \Delta_t(\mathbf{w}_t, \mathbf{w}_{t,\mathrm{ref}}). \tag{22}$$

Rearranging the terms in (21), dividing by $VT$ and subsequently summing over all tasks $t$, we get

$$\frac{1}{T}\sum_{t=1}^{T}[\Phi_t(\mathbf{w}_t) - \Phi_t(\mathbf{w}_t^*)] \quad \leq \quad \frac{1}{VT}\sum_{t=1}^{T}\Delta_t(\mathbf{w}_t, \mathbf{w}_{t,\mathrm{ref}}) - \frac{1}{VT}\sum_{t=1}^{T}\Delta\mathcal{L}[t]$$

$$\overset{(a)}{\leq} \quad \frac{1}{VT}\sum_{t=1}^{T}\Delta_t(\mathbf{w}_t, \mathbf{w}_{t,\mathrm{ref}}) - \frac{1}{VT}\left[\mathcal{L}_{\mathbf{w}_T}[T] - \mathcal{L}_{\mathbf{w}_0}[0]\right]$$

$$\overset{(b)}{\leq} \quad \frac{1}{VT}\sum_{t=1}^{T}\Delta_t(\mathbf{w}_t, \mathbf{w}_{t,\mathrm{ref}}), \tag{23}$$

where the last term on the right-hand side in $(a)$ is obtained by using the telescoping sum. The last inequality $(b)$ follows from the fact that $\mathcal{L}_{\mathbf{w}_0}[0] = 0$ and $\mathcal{L}_{\mathbf{w}_t}[T] \geq 0$. This completes the proof.

## C   Proof of Theorem 2

Consider the following

$$
\begin{aligned}
V\Phi_t(\mathbf{w}_t) + \Delta\mathcal{L}[t] &\leq V\Phi_t(\mathbf{w}_t) + \Delta_t(\mathbf{w}_t, \mathbf{w}_{t,\mathrm{ref}}) + f(\mathbf{w}_t, \mathbf{w}_{t,\mathrm{ref}}) \\
&\leq V\Phi_t(\mathbf{w}_t^*) + \Delta_t(\mathbf{w}_t, \mathbf{w}_{t,\mathrm{ref}}) + f(\mathbf{w}_t^*, \mathbf{w}_{t,\mathrm{ref}}), \\
&= V\Phi_t(\mathbf{w}_t^*) + \Delta_t(\mathbf{w}_t, \mathbf{w}_{t,\mathrm{ref}}) + \frac{1}{t-1}\sum_{k=1}^{t-1} Q_k[t-1]\Delta\hat{\Phi}_k(\mathbf{w}_t^*, \tilde{\mathbf{w}}_{t,k}), \\
&= V\Phi_t(\mathbf{w}_t^*) + \Delta_t(\mathbf{w}_t, \mathbf{w}_{t,\mathrm{ref}}) + \frac{1}{t-1}\sum_{k=1}^{t-1} Q_k[t-1](\hat{\Phi}_k(\mathbf{w}_t^*) - \hat{\Phi}_k(\tilde{\mathbf{w}}_{t,k}) - \delta), \\
&\leq V\Phi_t(\mathbf{w}_t^*) + \Delta_t(\mathbf{w}_t, \mathbf{w}_{t,\mathrm{ref}}) + \frac{1}{t-1}\sum_{k=1}^{t-1} Q_k[t-1](\hat{\Phi}_k(\mathbf{w}_t^*) - \inf_{\mathbf{v}\in\mathcal{W}} \hat{\Phi}_k(\mathbf{v}) - \delta), \quad (24)
\end{aligned}
$$

where the first inequality follows from the fact that $\mathbf{w}_t$ is an optimal solution to DPP function, and the last inequality holds since $\hat{\Phi}_k(\tilde{\mathbf{w}}_{t,k}) \geq \inf_{\mathbf{v}\in\mathcal{W}} \hat{\Phi}_k(\mathbf{v})$. Recall that $\hat{\Phi}_k(\mathbf{w}_t^*) - \inf_{\mathbf{v}\in\mathcal{W}} \hat{\Phi}_k(\mathbf{v}) - \delta \leq -\epsilon$, where $\epsilon := (\delta - \delta') > 0$ since $\delta > \delta'$. Using this in the above, we get

$$
V\Phi_t(\mathbf{w}_t) + \Delta\mathcal{L}[t] \leq V\Phi_t(\mathbf{w}_t^*) + \Delta_t(\mathbf{w}_t, \mathbf{w}_{t,\mathrm{ref}}) - \tfrac{\epsilon}{t-1}\sum_{k=1}^{t-1} Q_k[t-1]. \quad (25)
$$

After rearranging the terms above and dividing both sides by $\delta$, we obtain

$$
\begin{aligned}
\frac{1}{t-1}\sum_{k=1}^{t-1} Q_k[t-1] &\leq \frac{V}{\epsilon}[\Phi_t(\mathbf{w}_t^*) - \Phi_t(\mathbf{w}_t)] + \frac{\Delta_t(\mathbf{w}_t, \mathbf{w}_{t,\mathrm{ref}})}{\epsilon} - \frac{\Delta\mathcal{L}[t]}{\epsilon} \\
&= \frac{V}{\epsilon}[\hat{\Phi}_{t-1}(\mathbf{w}_t^*) - \hat{\Phi}_{t-1}(\mathbf{w}_t)] + \frac{V}{\epsilon}[\Phi_t(\mathbf{w}_t^*) - \hat{\Phi}_{t-1}(\mathbf{w}_t^*) + \hat{\Phi}_{t-1}(\mathbf{w}_t) - \Phi_t(\mathbf{w}_t)] \\
&\quad + \frac{\Delta_t(\mathbf{w}_t, \mathbf{w}_{t,\mathrm{ref}})}{\epsilon} - \frac{\Delta\mathcal{L}[t]}{\epsilon} \\
&\leq \frac{V}{\epsilon}[\hat{\Phi}_{t-1}(\mathbf{w}_t^*) - \inf_{\mathbf{u}} \hat{\Phi}_{t-1}(\mathbf{u})] + \frac{2V}{\epsilon}\sup_{\mathbf{w}}|\hat{\Phi}_{t-1}(\mathbf{w}) - \Phi_t(\mathbf{w})| + \frac{\Delta_t(\mathbf{w}_t, \mathbf{w}_{t,\mathrm{ref}})}{\epsilon} - \frac{\Delta\mathcal{L}[t]}{\epsilon} \\
&\leq \frac{V\delta'}{\epsilon} + \frac{2V}{\epsilon}\sup_{\mathbf{w}}|\hat{\Phi}_{t-1}(\mathbf{w}) - \Phi_t(\mathbf{w})| + \frac{\Delta_t(\mathbf{w}_t, \mathbf{w}_{t,\mathrm{ref}})}{\epsilon} - \frac{\Delta\mathcal{L}[t]}{\epsilon}, \quad (26)
\end{aligned}
$$

where the last inequality follows from the fact that the optimal sequence $\mathbf{w}_t^*$ is feasible for (2), and hence $\hat{\Phi}_{t-1}(\mathbf{w}_t^*) - \inf_{\mathbf{u}} \hat{\Phi}_{t-1}(\mathbf{u}) < \delta'$. After dropping the negative term and averaging over all tasks $t$, we get

$$
\frac{1}{T}\sum_{t=1}^{T} \bar{Q}[t-1] \;\leq\; \frac{V\delta'}{\epsilon} + \frac{1}{T\epsilon}\sum_{t=1}^{T}\Delta_t(\mathbf{w}_t, \mathbf{w}_{t,\mathrm{ref}}) + \frac{2V D_{\Phi,\hat{\Phi}}[T]}{\epsilon}, \quad (27)
$$

where $\bar{Q}[t-1] := \frac{1}{t-1}\sum_{k=1}^{t-1} Q_k[t-1]$, and $D_{\Phi,\hat{\Phi}}[T] = \frac{1}{T}\sum_{t=1}^{T}\sup_{\mathbf{w}}|\Phi_t(\mathbf{w}) - \hat{\Phi}_{t-1}(\mathbf{w})|$.

## D   Convergence of GD

In this section, we provide a proof of convergence guarantee of the GD for DPP. In the next section, we provide a proof of an upper bound on the average gradient squared.

## E   Proof of Theorem 3

Recall that the DPP term is given by $\mathtt{DPP}_{V,t}(\mathbf{w}) = V\Phi_t(\mathbf{w}) + \sum_{k=1}^{t-1}\Delta\hat{\Phi}_k(\mathbf{w}, \tilde{\mathbf{w}}_{t,k})$. The proposed algorithm performs GD on $\mathtt{DPP}_{V,t}(\mathbf{w})$. First, note that the DPP $\mathtt{DPP}_{V,t}(\mathbf{w})$ is $\alpha_t := VL + \frac{l}{t-1}\sum_{k=1}^{t-1} Q_k[t-1]$ smooth–a property proved in Lemma 2 in the main text. The smoothness leads to

$$
\mathtt{DPP}_{V,t}(\mathbf{w}_{t+1}) \leq \mathtt{DPP}_{V,t}(\mathbf{w}_t) + \eta\langle G_V(\mathbf{w}_t), \mathbf{w}_{t+1} - \mathbf{w}_t\rangle + \alpha_t\|\mathbf{w}_{t+1} - \mathbf{w}_t\|^2,
$$

where $G_V(\mathbf{w}_t) := \nabla \text{DPP}_{V,t}(\mathbf{w}_t)$. Substituting the update given in (14), we obtain

$$
\begin{aligned}
\text{DPP}_{V,t}(\mathbf{w}_{t+1}) &\leq \text{DPP}_{V,t}(\mathbf{w}_t) - \eta \|\nabla \text{DPP}_{V,t}(\mathbf{w}_t)\|^2 + \alpha_t \eta^2 \|\nabla \text{DPP}_{V,t}(\mathbf{w}_t)\|^2 \\
&= \text{DPP}_{V,t}(\mathbf{w}_t) - \eta(1 - \alpha_t \eta) \|\nabla \text{DPP}_{V,t}(\mathbf{w}_t)\|^2,
\end{aligned}
\tag{28}
$$

where $\eta \leq 1/2\alpha_t$ for all $t \in [T]$, which makes the second term in the above non-negative. Rearranging the terms, we obtain

$$
\|\nabla \text{DPP}_{V,t}(\mathbf{w}_t)\|^2 \leq \frac{2}{\eta}[\text{DPP}_{V,t}(\mathbf{w}_t) - \text{DPP}_{V,t}(\mathbf{w}_{t+1})].
\tag{29}
$$

Now, it remains to prove a bound on the drift difference above, which is the essence of the following lemma.

**Lemma 4.** *The average DPP across all tasks is bounded as follows*

$$
\frac{1}{T\eta} \sum_{t=1}^{T} [\textit{DPP}_{V,t}(\mathbf{w}_t) - \textit{DPP}_{V,t}(\mathbf{w}_{t+1})] \leq \frac{V\Phi_1(\mathbf{w}_1)}{T\eta} + \frac{V}{\eta T} \sum_{t=1}^{T} \sup_{\mathbf{w}} \left| \Phi_{t+1}(\mathbf{w}) - \Phi_t(\mathbf{w}) \right| + \frac{r^2 l}{2\eta T} \sum_{t=1}^{T} \bar{Q}_k[t-1].
$$

*Proof:* Consider the following

$$
\begin{aligned}
\frac{\text{DPP}_{V,t}(\mathbf{w}_t) - \text{DPP}_{V,t}(\mathbf{w}_{t+1})}{\eta} &= \frac{[V\Phi_t(\mathbf{w}_t) - V\Phi_t(\mathbf{w}_{t+1})]}{\eta} + \frac{1}{\eta(t-1)} \sum_{k=1}^{t-1} Q_k[t-1]\left[ \hat{\Phi}_k(\mathbf{w}_t) - \hat{\Phi}_k(\tilde{\mathbf{w}}_{t,k}) - \delta \right] \\
&\quad - \frac{1}{\eta(t-1)} \sum_{k=1}^{t-1} Q_k[t-1]\left[ \hat{\Phi}_k(\mathbf{w}_{t+1}) - \hat{\Phi}_k(\tilde{\mathbf{w}}_{t,k}) - \delta \right] \\
&= \frac{V}{\eta} \underbrace{\left[ \Phi_t(\mathbf{w}_t) - \Phi_t(\mathbf{w}_{t+1}) \right]}_{(a)} + \frac{1}{\eta(t-1)} \underbrace{\sum_{k=1}^{t-1} Q_k[t-1]\left[ \hat{\Phi}_k(\mathbf{w}_t) - \hat{\Phi}_k(\mathbf{w}_{t+1}) \right]}_{(b)} \quad (30)
\end{aligned}
$$

By adding and subtracting $\Phi_{t+1}(\mathbf{w}_{t+1})$ to the term $(a)$ above, we get

$$
\begin{aligned}
\Phi_t(\mathbf{w}_t) - \Phi_t(\mathbf{w}_{t+1}) &= \Phi_t(\mathbf{w}_t) - \Phi_t(\mathbf{w}_{t+1}) + \Phi_{t+1}(\mathbf{w}_{t+1}) - \Phi_{t+1}(\mathbf{w}_{t+1}) \\
&\leq \Phi_t(\mathbf{w}_t) - \Phi_{t+1}(\mathbf{w}_{t+1}) + \sup_{\mathbf{w}} |\Phi_{t+1}(\mathbf{w}) - \Phi_t(\mathbf{w})|
\end{aligned}
\tag{31}
$$

The second term $(b)$ can be written as

$$
\begin{aligned}
\sum_{k=1}^{t-1} Q_k[t-1]\left[ \hat{\Phi}_k(\mathbf{w}_t) - \hat{\Phi}_k(\mathbf{w}_{t+1}) \right] &\leq \sum_{k=1}^{t-1} Q_k[t-1]\left[ \hat{\Phi}_k(\mathbf{w}_t) - \hat{\Phi}_k(\mathbf{v}_k^*) + \hat{\Phi}_k(\mathbf{v}_k^*) - \hat{\Phi}_k(\mathbf{w}_{t+1}) \right] \\
&= \sum_{k=1}^{t-1} Q_k[t-1]\left[ \hat{\Phi}_k(\mathbf{w}_t) - \hat{\Phi}_k(\mathbf{v}_k^*) \right] + \sum_{k=1}^{t-1} Q_k[t-1]\left[ \hat{\Phi}_k(\mathbf{v}_k^*) - \hat{\Phi}_k(\mathbf{w}_{t+1}) \right] \\
&\leq \frac{r^2 l}{2} \sum_{k=1}^{t-1} Q_k(t-1) \\
&\leq \frac{r^2 l(t-1)}{2} \bar{Q}[t-1],
\end{aligned}
\tag{32}
$$

where we obtained the third expression using $\sum_{k=1}^{t-1} Q_k[t-1]\left[ \hat{\Phi}_k(\mathbf{v}_k^*) - \hat{\Phi}_k(\mathbf{w}_{t+1}) \right] < 0$ and Lemma 3. The last expression is obtained using the definition $\bar{Q}[t-1] = \frac{1}{(t-1)} \sum_{k=1}^{t-1} Q_k(t-1)$. Combining the terms above, we can now rewrite (30) as

$$
\frac{2(\text{DPP}_{V,t}(\mathbf{w}_t) - \text{DPP}_{V,t}(\mathbf{w}_{t+1}))}{\eta} \leq \frac{2V}{\eta}\left( \Phi_t(\mathbf{w}_t) - \Phi_{t+1}(\mathbf{w}_{t+1}) \right) + \frac{2V}{\eta} \sup_{\mathbf{w}} \left| \Phi_{t+1}(\mathbf{w}) - \Phi_t(\mathbf{w}) \right| + \frac{r^2 l}{\eta} \bar{Q}[t-1].
\tag{33}
$$

Now summing across all the tasks and dividing by $T$ gives the following

$$\frac{1}{\eta T}\sum_{t=1}^{T}[\mathrm{DPP}_{V,t}(\mathbf{w}_t) - \mathrm{DPP}_{V,t}(\mathbf{w}_{t+1})] \leq \frac{V\Phi_1(\mathbf{w}_1)}{T\eta} + \frac{V}{\eta T}\sum_{t=1}^{T}\sup_{\mathbf{w}}\left|\Phi_{t+1}(\mathbf{w}) - \Phi_t(\mathbf{w})\right| + \frac{r^2 l}{2\eta T}\sum_{t=1}^{T}\bar{Q}[t-1], \quad (34)$$

where we telescope over the first term, and neglected the negative term in the first term from the right hand side above. This results in the desired result of Lemma 4.

Now, let us lower bound the left hand side in (29). Towards this, using $\nabla\mathrm{DPP}_{V,t}(\mathbf{w}_t) = V\nabla\Phi_t(\mathbf{w}_t) + \sum_{k=1}^{t-1}\frac{Q_k(t-1)}{t-1}\nabla\hat{\Phi}_k(\mathbf{w}_t)$, and expanding the norm squared, we get

$$||\nabla\mathrm{DPP}_{V,t}(\mathbf{w}_t)||^2 = V^2||\nabla\Phi_t(\mathbf{w}_t)||^2 + \left\|\frac{1}{t-1}\sum_{k=1}^{t-1}Q_k(t-1)\nabla\hat{\Phi}_k(\mathbf{w}_t)\right\|^2 + 2V\sum_{k=1}^{t-1}\frac{Q_k(t-1)}{t-1}\langle\nabla\Phi_t(\mathbf{w}_t), \nabla\hat{\Phi}_k(\mathbf{w}_t)\rangle$$

$$\geq V^2||\nabla\Phi_t(\mathbf{w}_t)||^2 + 2V\sum_{k=1}^{t-1}\frac{Q_k(t-1)}{t-1}\langle\nabla\Phi_t(\mathbf{w}_t), \nabla\hat{\Phi}_k(\mathbf{w}_t)\rangle$$

$$\geq V^2||\nabla\Phi_t[\mathbf{w}_t]||^2 + 2V\sum_{k=1}^{t-1}\frac{Q_k[t-1]}{t-1}\rho_{t,k}$$

$$\geq V^2||\nabla\Phi_t[\mathbf{w}_t]||^2 + 2V\sum_{k=1}^{t-1}\frac{Q_k[t-1]}{t-1}\rho_{t,k}\mathbf{1}\{\rho_{t,k}\leq 0\}$$

$$\geq V^2||\nabla\Phi_t[\mathbf{w}_t]||^2 - Vlr^2\rho_{t,-}, \quad (35)$$

where the last inequality follows from substituting the queue bound, and using $\rho_{t,k} := \inf_{\mathbf{w}\in\mathcal{W}}\langle\nabla\Phi_t(\mathbf{w}), \nabla\hat{\Phi}_k(\mathbf{w})\rangle$. Further, we substitute the $\rho_{t,-} := -\sum_{k=1}^{t-1}\rho_{t,k}\mathbf{1}\{\rho_{t,k}\leq 0\}$ to obtain the last inequality above. Rearranging and averaging across all tasks and dividing by $V^2$, we get

$$\frac{1}{T}\sum_{t=1}^{T}||\nabla\Phi_t(\mathbf{w}_t)||^2 \leq \frac{1}{TV^2}\sum_{t=1}^{T}||\nabla\mathrm{DPP}_{V,t}(\mathbf{w})||^2 + \frac{lr^2}{TV}\sum_{t=1}^{T}\bar{\rho}_{t,-}. \quad (36)$$

Using 29 and Lemma 4, we obtain the following

$$\frac{1}{T}\sum_{t=1}^{T}||\nabla\Phi_t(\mathbf{w}_t)||^2 \leq \frac{2\Phi_1(\mathbf{w}_1)}{TV\eta} + \frac{2}{\eta TV}\sum_{t=1}^{T}\sup_{\mathbf{w}}\left|\Phi_{t+1}(\mathbf{w}) - \Phi_t(\mathbf{w})\right| + \frac{lr^2}{TV}\sum_{t=1}^{T}\bar{\rho}_{t,-} + \frac{r^2 l}{\eta TV^2}\sum_{t=1}^{T}\bar{Q}[t-1]. \quad (37)$$

## F  Proof of Theorem 4

As in the proof of Theorem 2, consider the following with $\tilde{\mathbf{w}}_{t,k} = \mathbf{w}_{t-1}$ for all $k$ as in Case 2:

$$V\Phi_t(\mathbf{w}_t) + \Delta\mathcal{L}[t] \leq V\Phi_t(\mathbf{w}_t) + \Delta_t(\mathbf{w}_t, \mathbf{w}_{t-1}) + f(\mathbf{w}_t, \mathbf{w}_{t-1})$$

$$\overset{(a)}{\leq} V\Phi_t(\mathbf{w}_{t-1}) + \Delta_t(\mathbf{w}_t, \mathbf{w}_{t-1}) + f(\mathbf{w}_{t-1}, \mathbf{w}_{t-1}),$$

$$= V\Phi_t(\mathbf{w}_{t-1}) + \Delta_t(\mathbf{w}_t, \mathbf{w}_{t-1}) + \frac{1}{t-1}\sum_{k=1}^{t-1}Q_k[t-1]\Delta\hat{\Phi}_k(\mathbf{w}_{t-1}, \mathbf{w}_{t-1}),$$

$$\leq V\Phi_t(\mathbf{w}_{t-1}) + \Delta_t(\mathbf{w}_t, \mathbf{w}_{t-1}) - \frac{\delta}{t-1}\sum_{k=1}^{t-1}Q_k[t-1], \quad (38)$$

where $(a)$ follows from the fact that $\mathbf{w}_t$ is obtained by using one step GD starting from $\mathbf{w}_{t-1}$, which results in a smaller DPP, the last inequality follows from the fact that $\mathbf{w}_t$ is an optimal solution to DPP function.

Recall that $\Delta\hat{\Phi}_k(\mathbf{w}_{t-1}, \mathbf{w}_{t-1}) = -\delta$. Rearranging the terms above and dividing both sides by $\delta$, we obtain

$$\frac{1}{t-1}\sum_{k=1}^{t-1} Q_k[t-1] \leq \frac{V}{\delta}[\Phi_t(\mathbf{w}_{t-1}) - \Phi_t(\mathbf{w}_t)] + \frac{\Delta_t(\mathbf{w}_t, \mathbf{w}_{t-1})}{\delta} - \frac{\Delta\mathcal{L}[t]}{\delta}$$

$$= \frac{V}{\delta}[\Phi_t(\mathbf{w}_{t-1}) - \Phi_{t-1}(\mathbf{w}_{t-1}) + \Phi_{t-1}(\mathbf{w}_{t-1}) - \Phi_t(\mathbf{w}_t)] + \frac{\Delta_t(\mathbf{w}_t, \mathbf{w}_{t-1})}{\delta} - \frac{\Delta\mathcal{L}[t]}{\delta}$$

$$\leq \frac{V}{\delta}\sup_{\mathbf{w}}|\Phi_t(\mathbf{w}) - \Phi_{t-1}(\mathbf{w})| + \frac{V}{\delta}(\Phi_{t-1}(\mathbf{w}_{t-1}) - \Phi_t(\mathbf{w}_t)) + \frac{\Delta_t(\mathbf{w}_t, \mathbf{w}_{t-1})}{\delta} - \frac{\Delta\mathcal{L}[t]}{\delta}. \quad (39)$$

Note that the terms $\sup_{\mathbf{w}}|\Phi_t(\mathbf{w}) - \Phi_{t-1}(\mathbf{w})|$ and $(\Phi_{t-1}(\mathbf{w}_{t-1}) - \Phi_t(\mathbf{w}_t))$ are bounded since the functions are bounded. Similarly, the last two terms are also bounded, and hence $\frac{1}{t-1}\sum_{k=1}^{t-1} Q_k[t-1]$ scales as $\mathcal{O}(V)$. This completes the proof of the first result of the theorem. Now, averaging the upper bound in (39) over all tasks $t$ results in

$$\frac{1}{T}\sum_{t=1}^{T}\bar{Q}[t-1] \leq \frac{V\mathsf{D}_\Phi[T]}{\delta} + \frac{1}{T\delta}\sum_{t=1}^{T}\Delta_t(\mathbf{w}_t, \mathbf{w}_{t-1}) - \frac{1}{T}[\mathcal{L}_{\mathbf{w}_T}[T] - \mathcal{L}_{\mathbf{w}_0}[0]],$$

$$\leq \frac{V\mathsf{D}_\Phi[T]}{\delta} + \frac{1}{T\delta}\sum_{t=1}^{T}\Delta_t(\mathbf{w}_t, \mathbf{w}_{t-1}), \quad (40)$$

where $\bar{Q}[t-1] := \frac{1}{t-1}\sum_{k=1}^{t-1} Q_k[t-1]$, and $\mathsf{D}_\Phi[T] = \frac{1}{T}\sum_{t=1}^{T}\sup_{\mathbf{w}}|\Phi_t(\mathbf{w}) - \Phi_{t-1}(\mathbf{w})|$. Further, in the first inequality, we have used the fact that summing $(\Phi_{t-1}(\mathbf{w}_{t-1}) - \Phi_t(\mathbf{w}_t))$ over $t$ results in $\Phi_0(\mathbf{w}_0) - \Phi_T(\mathbf{w}_T) = -\Phi_T(\mathbf{w}_T) \leq 0$. While telescoping the last term, we have used the fact that $\hat{\Phi}_0(\mathbf{w}_0) = 0$, and the last inequality follows from ignoring the negative term. $\qquad\square$

## G   Additional Experimental Results

Here, we present hyperparameter settings, implementation settings, and results that support and extend the empirical findings reported in the main paper.

### G.1   Dataset Details

In the main paper, we have presented a brief review of the datasets used. Here, we provide a more detailed description. We report experimental results on standard CL benchmarks Van de Ven et al. (2022), including Permuted MNIST (PMNIST), Split-MNIST, Split-CIFAR10, Split-CIFAR100, and Split-TinyImageNet. PMNIST is a variant of the MNIST dataset Deng (2012), where each task applies a fixed random permutation to the input pixels, resulting in a sequence of tasks with distinct input distributions while preserving label semantics. Split-MNIST is constructed by partitioning the MNIST dataset into multiple disjoint tasks, each containing a subset of digit classes, with the class labels remaining consistent across tasks. Split-CIFAR10 is derived from the CIFAR10 dataset Krizhevsky & Hinton (2009) by dividing the ten classes into 5 disjoint tasks, each containing 2 classes. Similarly, Split-CIFAR100 is obtained from CIFAR100 Krizhevsky & Hinton (2009), where the dataset is split into 5 tasks, each comprising 20 classes. Finally, Split-TinyImageNet is constructed from the TinyImageNet dataset Le & Yang (2015) by dividing the data into 20 tasks, each consisting of 10 classes.

### G.2   Baselines

We compare the proposed method in the task-incremental setting with the following baselines. In the task-incremental setup, task identity is required during both the training and testing phases. Further, all classifier heads are initialized as a list at the beginning of training and selected based on the task identity for training and testing. We compare our method with several baselines as follows: (a) `Fine-tune`, which sequentially trains on tasks without any mechanism to prevent forgetting; (b) `EWC` Kirkpatrick et al. (2017), a Fisher information–based regularization method; (c) `A-GEM` Chaudhry et al. (2018), a gradient projection-based

Table 2: Hyperparameters used in `COLD-ORACLE` experiments.

| Hyperparameter | Permuted-MNIST | Split-CIFAR10 | | Split-CIFAR100 | | Split-TinyImageNet | |
| --- | --- | --- | --- | --- | --- | --- | --- |
| | $|\mathcal{M}|$=5000 | $|\mathcal{M}|$=600 | $|\mathcal{M}|$=1000 | $|\mathcal{M}|$=1000 | $|\mathcal{M}|$=5000 | $|\mathcal{M}|$=2000 | $|\mathcal{M}|$=5000 |
| Learning rate | 0.0001 | 0.0003 | 0.0001 | 0.0003 | 0.0003 | 0.0001 | 0.0001 |
| Optimizer | ADAM | ADAM | SGD-M | ADAM | ADAM | SGD-M | SGD-M |
| Epochs (default) | 5 | 5 | 5 | 5 | 5 | 5 | 5 |
| # Tasks (default) | 20 | 5 | 5 | 20 | 20 | 20 | 20 |
| $V$ | 20 | 5 | 100 | 5 | 40 | 1000 | 1000 |
| Queue initialization | 0 | 0 | 0 | 0 | 0 | 0 | 0 |
| $\delta$-parameter | 4 | 0.3 | 0.9 | 0.5 | 0.9 | 0.5 | 0.9 |
| Task batch size | 128 | 10 | 64 | 10 | 64 | 64 | 64 |
| Memory batch size | 1 | 7 | 7 | 7 | 7 | 7 | 7 |

Table 3: Hyperparameters used in `COLD` experiments.

| Hyperparameter | Permuted-MNIST | Split-CIFAR10 | | Split-CIFAR100 | | Split-TinyImageNet | |
| --- | --- | --- | --- | --- | --- | --- | --- |
| | $|\mathcal{M}|$=5000 | $|\mathcal{M}|$=600 | $|\mathcal{M}|$=1000 | $|\mathcal{M}|$=1000 | $|\mathcal{M}|$=5000 | $|\mathcal{M}|$=2000 | $|\mathcal{M}|$=5000 |
| Learning rate | 0.0001 | 0.0003 | 0.0003 | 0.0003 | 0.0003 | 0.0001 | 0.0001 |
| Optimizer | ADAM | ADAM | ADAM | ADAM | ADAM | SGD-M | SGD-M |
| Epochs (default) | 5 | 5 | 5 | 5 | 5 | 5 | 5 |
| # Tasks (default) | 20 | 5 | 5 | 20 | 20 | 20 | 20 |
| $V$ | 20 | 5 | 90 | 7 | 10 | 200 | 200 |
| Queue initialization | 0 | 0 | 0 | 0 | 0 | 0 | 0 |
| $\delta$-parameter | 2 | 0.3 | 0.5 | 2 | 1 | 2 | 2 |
| Task batch size | 128 | 10 | 10 | 10 | 64 | 64 | 64 |
| Memory batch size | 1 | 7 | 7 | 7 | 7 | 7 | 7 |

replay method; (d) `NCCL` Han et al. (2023), is a memory-based continual method that uses adaptive step sizes for current and previously seen tasks. (e) `MER` Riemer et al. (2019), a meta-learning–based replay method; (f) `GDumb` Prabhu et al. (2020), which trains only on the replay buffer at evaluation time; (g) `DER` and `DER++` Buzzega et al. (2020), which combine experience replay with logit-based distillation; (i) `ER-ACE` Caccia et al. (2021), an experience replay method with adaptive class balancing; (j) `CBA` Wang et al. (2023), a bias adaptation module that alleviates catastrophic distribution shifts; and (k) `REFRESH` Wang et al. (2024), a recent replay-based approach designed to improve stability.[7] For the domain-incremental setting, we compare the proposed method against a relevant subset of these baselines that are applicable to this scenario.

## G.3 Hyperparameter Settings

Table. 2 and 3 list the default hyperparameter settings adopted in our experiments on Permuted-MNIST, Split-CIFAR10, Split-CIFAR100, and Split-TinyImageNet. These values were used consistently unless varied explicitly for specific experiments. All experiments were conducted on NVIDIA RTX A5000 (24GB) and NVIDIA L40 (46GB) GPUs. We adopt the hyperparameters reported in the original works when the experimental settings align with ours. When configurations are not directly applicable, we have tuned the methods over standard ranges using validation performance. We ensure fairness by allocating a comparable computational budget for hyperparameter search across all methods, including our own.

## G.4 Additional experiments

**Domain Incremental Setting:** From Table 4 and Fig.9, we observe that both `COLD-ORACLE` and `COLD` significantly outperform all baselines in the domain-incremental setting on PMNIST. In particular, the proposed methods achieve the highest average accuracy (above 91%) while simultaneously exhibiting the

---

[7]We use the Mammoth library Buzzega et al. (2020) and the Avalanche library Lomonaco et al. (2021) to reproduce baseline results. For `NCCL`, we report results from our own implementation, as the official code is not publicly available.

Table 4: **Domain-Incremental:** Performance on PMNIST ($M = 5$k).

| Method | Accuracy(%) | Forgetting |
|---|---|---|
| Fine-tune | $36.79 \pm 1.16$ | $0.64 \pm 0.01$ |
| EWC-2017 | $69.49 \pm 1.40$ | $0.26 \pm 0.03$ |
| A-GEM-2019 | $62.12 \pm 1.34$ | $0.34 \pm 0.005$ |
| DER-2020 | $78.72 \pm 0.14$ | $0.10 \pm 0.0008$ |
| DER++-2020 | $78.74 \pm 0.37$ | $0.06 \pm 0.0005$ |
| NCCL-2023 | $61.93 \pm 1.58$ | $0.36 \pm 0.01$ |
| **COLD (Ours)** | $\mathbf{91.26 \pm 0.22}$ | $\mathbf{0.04 \pm 0.003}$ |
| **COLD-ORACLE (Ours)** | $\mathbf{91.19 \pm 0.21}$ | $\mathbf{0.04 \pm 0.002}$ |

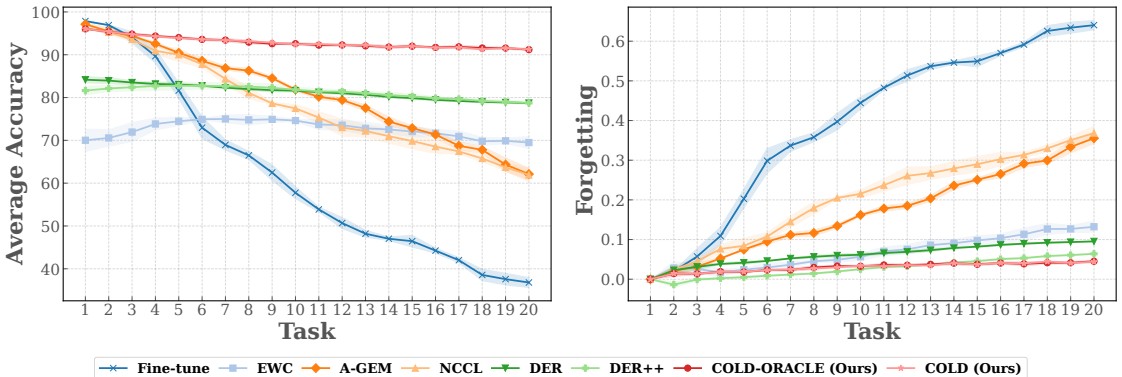

Figure 9: Comparison with baselines on PMNIST: average accuracy and forgetting across tasks.

Table 5: **Task-Incremental:** Performance on Split-CIFAR10 ($M = 1$k), Split-CIFAR100 ($M = 5$k), and Split-TinyImageNet ($M = 5$k).

| Method | Split-CIFAR10 | | Split-CIFAR100 | | Split-TinyImageNet | |
|---|---|---|---|---|---|---|
| | Accuracy (%) | Forgetting | Accuracy (%) | Forgetting | Accuracy (%) | Forgetting |
| Fine-tune | $76.70 \pm 5.31$ | $0.22 \pm 0.06$ | $29.86 \pm 1.54$ | $0.54 \pm 0.02$ | $16.59 \pm 0.86$ | $0.44 \pm 0.01$ |
| EWC-2017 | $71.05 \pm 4.72$ | $0.35 \pm 0.04$ | $35.00 \pm 2.76$ | $0.17 \pm 0.004$ | $24.12 \pm 0.75$ | $0.12 \pm 0.002$ |
| A-GEM-2019 | $83.81 \pm 4.24$ | $0.43 \pm 0.02$ | $69.82 \pm 3.36$ | $0.33 \pm 0.01$ | $53.43 \pm 0.10$ | $0.25 \pm 0.0005$ |
| MER-2019 | $76.91 \pm 0.43$ | $0.02 \pm 0.02$ | $49.68 \pm 2.14$ | $0.05 \pm 0.02$ | $53.76 \pm 0.38$ | $0.004 \pm 0.002$ |
| GDumb-2020 | $63.01 \pm 2.63$ | $-$ | $51.42 \pm 4.72$ | $-$ | $57.34 \pm 0.73$ | $-$ |
| DER-2020 | $85.05 \pm 0.70$ | $0.04 \pm 0.001$ | $48.31 \pm 0.50$ | $0.24 \pm 0.004$ | $30.58 \pm 0.29$ | $0.18 \pm 0.16$ |
| DER++-2020 | $85.80 \pm 0.92$ | $0.03 \pm 0.005$ | $71.67 \pm 1.25$ | $0.04 \pm 0.007$ | $58.88 \pm 0.55$ | $0.04 \pm 0.002$ |
| ER-ACE-2022 | $86.11 \pm 0.27$ | $0.02 \pm 0.01$ | $70.14 \pm 1.36$ | $0.04 \pm 0.01$ | $55.67 \pm 2.81$ | $0.05 \pm 0.02$ |
| CBA-2023 | $84.53 \pm 0.44$ | $0.006 \pm 0.001$ | $73.75 \pm 2.20$ | $0.02 \pm 0.01$ | $58.92 \pm 2.47$ | $0.03 \pm 0.01$ |
| NCCL-2023 | $71.90 \pm 1.29$ | $0.24 \pm 0.01$ | $50.79 \pm 1.38$ | $0.35 \pm 0.02$ | $30.15 \pm 1.65$ | $0.33 \pm 0.02$ |
| REFRESH-2024 | $85.87 \pm 1.87$ | $0.03 \pm 0.02$ | $70.84 \pm 4.67$ | $0.04 \pm 0.05$ | $60.08 \pm 0.99$ | $0.03 \pm 0.002$ |
| COLD (Ours) | $85.90 \pm 2.62$ | $0.09 \pm 0.01$ | $74.56 \pm 0.77$ | $0.05 \pm 0.01$ | $61.39 \pm 2.02$ | $0.07 \pm 0.0009$ |
| **COLD-ORACLE (Ours)** | $\mathbf{87.94 \pm 1.28}$ | $0.05 \pm 0.006$ | $\mathbf{75.64 \pm 1.11}$ | $0.05 \pm 0.01$ | $\mathbf{62.21 \pm 1.04}$ | $\mathbf{0.02 \pm 0.02}$ |

lowest forgetting among all compared approaches. This indicates that the proposed DPP-based replay strategy is highly effective at preserving previously learned knowledge under domain shifts. Notably, while replay-based baselines such as DER and DER++ improve over regularization and gradient-projection methods, they still lag behind the proposed methods by a substantial margin in both accuracy and forgetting.

**Comparison with Baselines:** We conduct experiments to benchmark the performance of the proposed COLD-ORACLE and COLD approaches against the baselines listed above. Tables 5 and 6 present a full tabular

Table 6: **Task-Incremental:** Performance on Split-CIFAR10 ($M = 600$), Split-CIFAR100 ($M = 1k$), and Split-TinyImageNet ($M = 2k$).

| Method | Split-CIFAR10 | | Split-CIFAR100 | | Split-TinyImageNet | |
|---|---|---|---|---|---|---|
| | Accuracy (%) | Forgetting | Accuracy (%) | Forgetting | Accuracy (%) | Forgetting |
| Fine-tune | $76.70 \pm 5.31$ | $0.22 \pm 0.06$ | $29.86 \pm 1.54$ | $0.54 \pm 0.02$ | $16.59 \pm 0.86$ | $0.44 \pm 0.01$ |
| EWC-2017 | $71.05 \pm 4.72$ | $0.35 \pm 0.04$ | $35.00 \pm 2.76$ | $0.17 \pm 0.004$ | $24.12 \pm 0.75$ | $0.12 \pm 0.002$ |
| A-GEM-2019 | $83.75 \pm 3.68$ | $0.42 \pm 0.02$ | $63.59 \pm 3.74$ | $0.30 \pm 0.006$ | $51.11 \pm 1.00$ | $0.23 \pm 0.004$ |
| MER-2019 | $77.32 \pm 2.72$ | $0.04 \pm 0.02$ | $47.35 \pm 1.17$ | $0.05 \pm 0.001$ | $34.89 \pm 0.45$ | $0.05 \pm 0.001$ |
| GDumb-2020 | $64.68 \pm 6.24$ | $-$ | $37.56 \pm 1.84$ | $-$ | $21.00 \pm 0.39$ | $-$ |
| DER-2020 | $83.66 \pm 0.93$ | $0.05 \pm 0.02$ | $50.27 \pm 1.38$ | $0.22 \pm 0.01$ | $28.22 \pm 0.86$ | $0.20 \pm 0.18$ |
| DER++-2020 | $84.84 \pm 1.34$ | $0.03 \pm 0.01$ | $66.96 \pm 0.91$ | $0.07 \pm 0.004$ | $56.19 \pm 1.00$ | $0.05 \pm 0.01$ |
| ER-ACE-2022 | $82.70 \pm 1.17$ | $0.03 \pm 0.02$ | $64.97 \pm 2.11$ | $0.05 \pm 0.04$ | $51.37 \pm 1.33$ | $0.07 \pm 0.01$ |
| CBA-2023 | $85.01 \pm 0.77$ | $0.02 \pm 0.008$ | $63.70 \pm 2.54$ | $0.03 \pm 0.01$ | $55.92 \pm 1.99$ | $0.04 \pm 0.01$ |
| NCCL-2023 | $72.99 \pm 7.93$ | $0.25 \pm 0.09$ | $29.20 \pm 1.45$ | $0.54 \pm 0.02$ | $19.64 \pm 1.38$ | $0.44 \pm 0.01$ |
| REFRESH-2024 | $84.93 \pm 1.23$ | $0.03 \pm 0.01$ | $67.34 \pm 1.14$ | $0.06 \pm 0.01$ | $56.68 \pm 2.03$ | $0.06 \pm 0.01$ |
| **COLD (Ours)** | $84.90 \pm 0.22$ | $0.07 \pm 0.05$ | $\mathbf{68.03 \pm 1.63}$ | $0.10 \pm 0.02$ | $55.09 \pm 1.53$ | $0.10 \pm 0.01$ |
| **COLD-ORACLE (Ours)** | $\mathbf{85.68 \pm 3.42}$ | $0.06 \pm 0.03$ | $67.38 \pm 0.02$ | $0.18 \pm 0.02$ | $\mathbf{56.74 \pm 1.35}$ | $0.08 \pm 0.008$ |

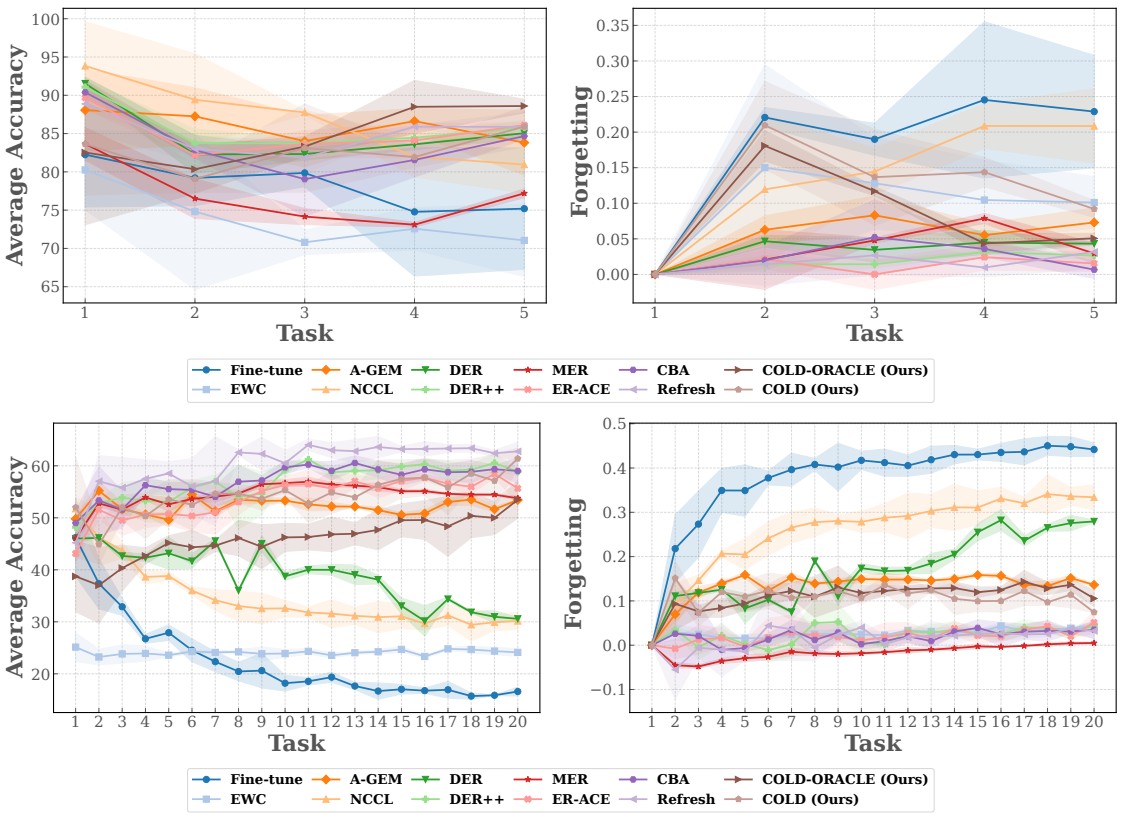

Figure 10: Comparison with the baselines: Average accuracy and forgetting across tasks. **Top:** Split-CIFAR10 and **Bottom:** Split-TinyImageNet.

comparison, analogous to the tables in the main paper, with all results averaged over three random seeds and corresponding standard deviation values reported. From Fig. 10 (**Top** and **Bottom**), we observe that the proposed methods outperform most baselines on the Split-TinyImageNet dataset with respect to the chosen evaluation metrics. In addition, the method can be explicitly tuned to reduce forgetting via the

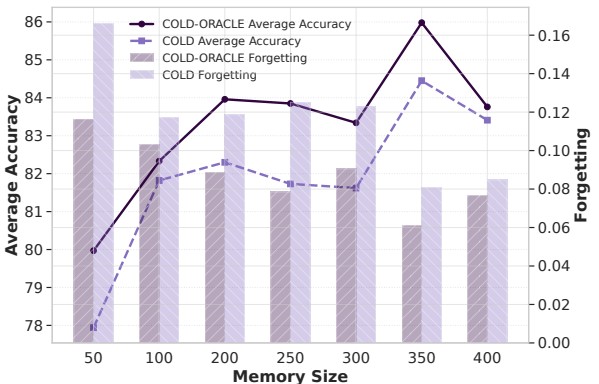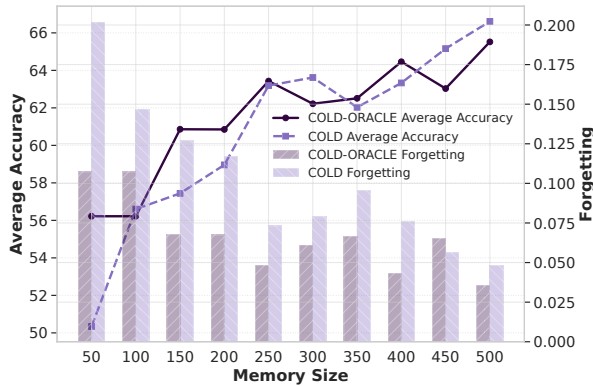

Figure 11: Effect of memory size $M$. **Left:** Split-CIFAR10. **Right:** Split-TinyImageNet. Results are shown for `COLD-ORACLE` and `COLD`.

algorithmic hyperparameters, allowing improved trade-offs between accuracy and forgetting. All plotted curves correspond to runs over three seeds.

In the following sections, we provide additional ablation studies by varying memory sampling sizes, the trade-off between accuracy and catastrophic forgetting with respect to the parameter $V$, varying $\delta$, number of epochs, and number of tasks. Selected results are provided in the main paper. **Note:** The key difference between `COLD-ORACLE` and `COLD` lie in the choice of reference model and how past-task constraints are enforced. `COLD-ORACLE` uses the best historical model for each task, leading to a stricter, low-variance constraint, whereas `COLD` uses the previous model, resulting in a weaker, replay-based constraint. In `COLD`, the replay-based gradient satisfies

$$\nabla\hat{\Phi}^{\mathrm{replay}}(w_t) = \nabla\Phi(w_t) + \epsilon_M, \quad \mathbb{E}\|\epsilon_M\|^2 = O\left(\frac{1}{M}\right),$$

which introduces estimation noise due to finite buffer size $M$. When $M$ is small, the error $\epsilon_M$ is large, so the replay loss is a noisy estimate of the true loss. In this regime, COLD's best-model reference provides a genuine advantage by enforcing tighter constraints through the virtual queue.

As $M$ increases, $\epsilon_M \to 0$, meaning the replay-based estimate becomes accurate and the empirical loss surface closely matches the true past-task loss. Consequently, all past models become approximately equivalent, and the best historical model and the previous model impose nearly identical constraints in the queue update. Thus, the performance gap diminishes because the choice of reference model becomes indistinguishable when the buffer provides a faithful approximation of past task distributions.

**Varying memory sampling size:** We analyze the effect of varying sizes of the memory samples per task in the proposed framework, using varying sampling sizes of 50, 100, 150, 200, 250, 300, 350, and 400 per task. We observe from Fig. 11 that for both Split-CIFAR10 and Split-TinyImageNet, average accuracy increases as the sampling size increases while the forgetting decreases, as expected. This trend is similar to what was observed in Split-CIFAR100.

**Varying Number of Epochs:** We study the effect of the number of SGD epochs per task on retention, forgetting, and adaptability in CL. In particular, when we perform too few epochs on the current task, the model underfits, leading to poor task-specific performance. On the other hand, if we perform too many epochs on the current task, the model overfits and aggressively adapts to the current task, increasing catastrophic forgetting of previous tasks. This is validated in Fig. 13 (**Left**) on the Split-TinyImageNet dataset, where we observe that with 5 epochs, it results in the best average accuracy ( 65%) and forgetting (0.04 − 0.06) combination on both the proposed algorithms. Increasing the epoch further results in higher forgetting, as depicted in the figure. Fig. 13 (**Right**) illustrates the average accuracy and forgetting gaps across tasks when comparing the online setting (1 epoch) with smaller batch settings. The batch setting achieves higher

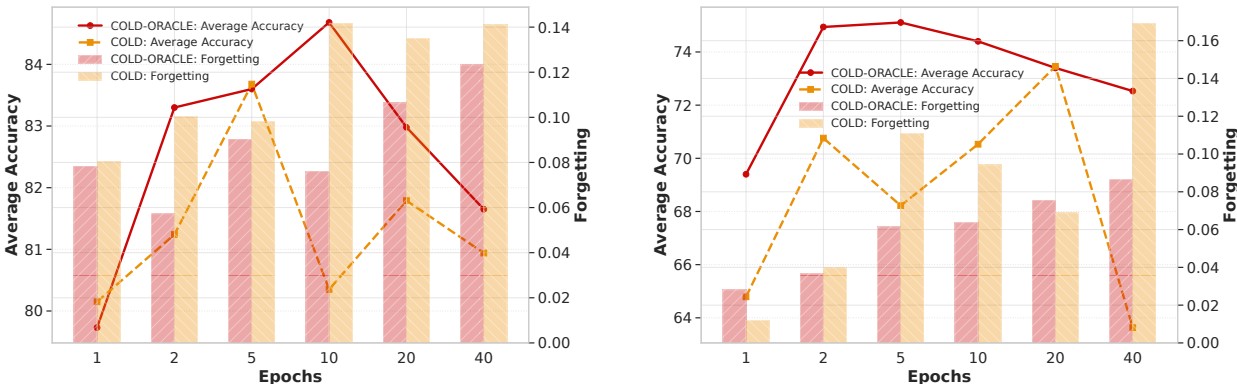

Figure 12: Effect of epochs per task. **Left:** Split-CIFAR10. **Right:** Split-CIFAR100. Results are shown for `COLD-ORACLE` and `COLD`.

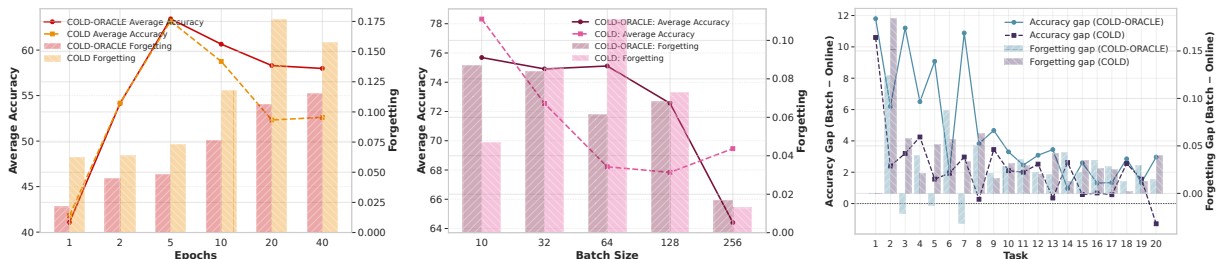

Figure 13: **Left:** `COLD-ORACLE` and `COLD` Algorithm: Effect of varying epochs per task on Split-TinyimageNet, **Center:** Varying batch size on Split-CIFAR100, and **Right:** Average accuracy gap and forgetting gap of online and batch settings on Split-CIFAR100

accuracy, as reflected by the positive accuracy gap. In contrast, the online setting exhibits lower forgetting, indicated by the positive forgetting gap in the figure. This reduced forgetting arises because, in the online setting, the model does not sufficiently learn the current task, which in turn leads to lower overall accuracy. Additionally, `COLD` consistently shows a smaller accuracy gap as compared to `COLD-ORACLE`.

In Fig. 13 (**Center**), we observe that small batch sizes result in higher final accuracy with little forgetting, while larger batch sizes lead to degraded performance. This can be attributed to the stochasticity of SGD, where noisy gradients with smaller batches act as implicit regularizers, helping the average accuracy across tasks, while larger batches approximate the full gradient descent, converging to sharper minima that fail to generalize across tasks Keskar et al. (2017). From a replay perspective, larger batch sizes result in fewer replay updates, which in turn reduces the reinforcement of earlier task knowledge. Similar observations can be made from Fig. 12 (**Right**), for the Split CIFAR10 and Split CIFAR100 datasets.

**Trade-off between Accuracy and Forgetting - Varying $V$:** Recall that in the proposed DPP problem, $V$ acts as a scaling factor on the current task relative to the constraints that abate forgetting. On the one hand, small values of $V$ prioritize stability, leading to lower values of forgetting, but may under-fit the current task. On the other hand, large values of $V$ prioritize the accuracy of current tasks, but increase forgetting. In Fig. 14, 15 we analyze the effect of varying the parameter $V$ on the Split-CIFAR10 and Split-TinyImageNet datasets, and we observe similar trends to Split-CIFAR100 (reported in the main manuscript). This also corroborates our theoretical findings.

**Effect of Varying Batchsize:** From Fig. 16, we observe that increasing the batch size reduces the average accuracy on both Split-CIFAR10 and Split-TinyImageNet. This trend is similar to Split-CIFAR100 in Fig. 13 (Center).

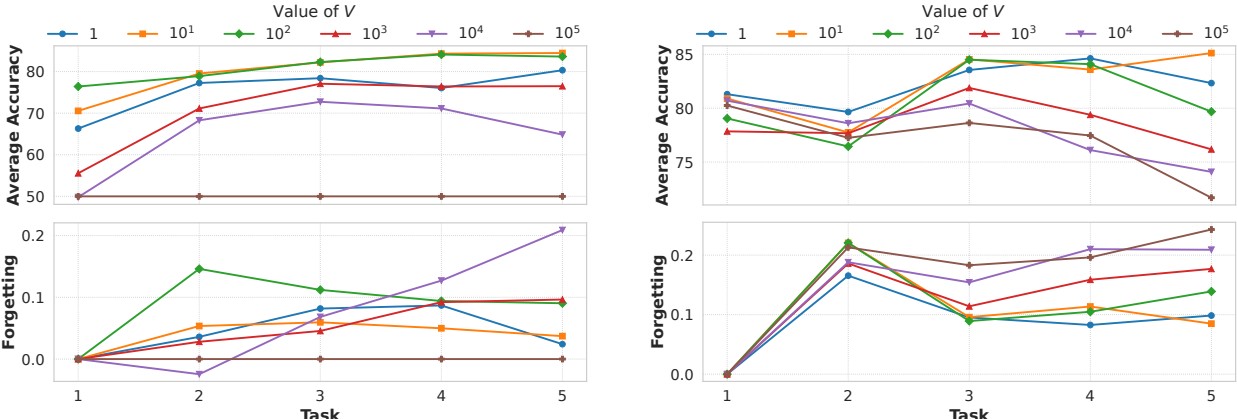

Figure 14: Trade-off between average accuracy and forgetting with varying $V$ on Split-CIFAR10. **Left:** `COLD-ORACLE`. **Right:** `COLD`.

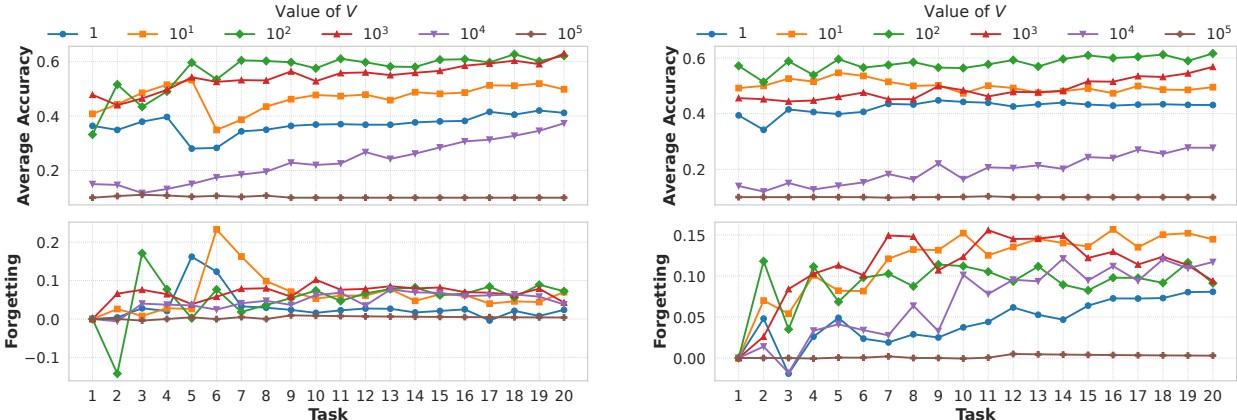

Figure 15: Trade-off between average accuracy and forgetting with varying $V$ on Split-TinyImageNet. **Left:** `COLD-ORACLE`. **Right:** `COLD`.

**Varying Number of Tasks:** In addition to Split-CIFAR100 (reported in the main manuscript), we analyze the effect of increasing the number of tasks on Split-TinyImageNet. As shown in Fig. 17, the experiment on Split-TinyImagenet shows similar trends to Split-CIFAR100, which confirms that the proposed method is well-suited for task-incremental settings.

**Online versus Batch Setting:** In Fig. 18, 19 we observe that the online setting for Split-CIFAR10 yields slightly lower accuracy compared to batch training but helps control forgetting, reflecting the benefit of lightweight updates in a more homogeneous dataset. We also observe a trade-off, i.e., online updates favor stability (lower forgetting), while batch updates favor plasticity (higher average accuracy). In the case of Split-TinyImageNet, the gap between online and batch average accuracy is wider.

**Queue Stability Analysis:** Empirical queue stability analysis in `COLD-ORACLE` and `COLD` validates whether the theoretical guarantees on constraint satisfaction hold in practice over a large number of tasks. In particular, if we observe that queue values are stable in practice, this validates the theoretical long-term guarantees as in Theorem 2. In contrast, if the queue values are unbounded, it indicates persistent constraint violations leading to catastrophic forgetting. In Fig. 20, for Split-TinyImageNet we observe that as $V$ increases, `COLD-ORACLE` prioritizes minimizing the new task loss more heavily. It becomes more plastic and it allows more violation of past constraints as reflected in the virtual queue, leading to a slower convergence towards regions of low forgetting.

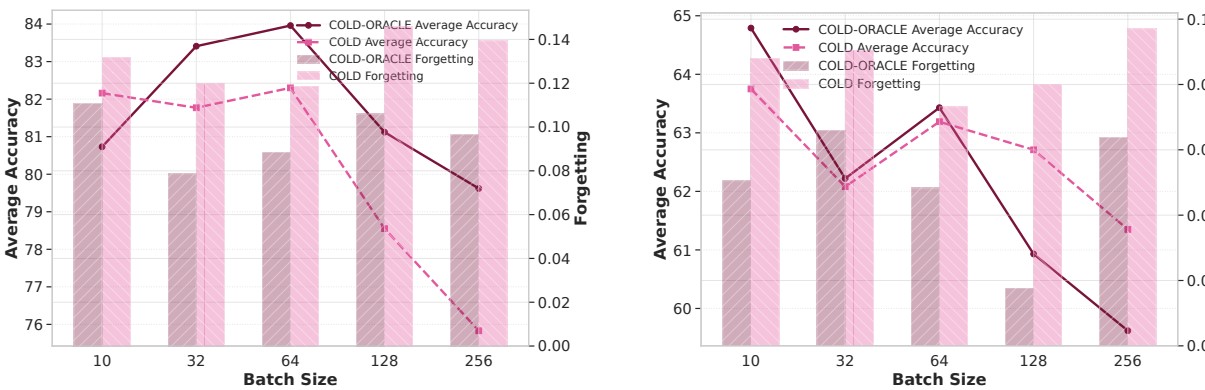

Figure 16: Effect of batch size in the offline setting. **Left:** Split-CIFAR10. **Right:** Split-TinyImageNet. Results are shown for `COLD-ORACLE` and `COLD`.

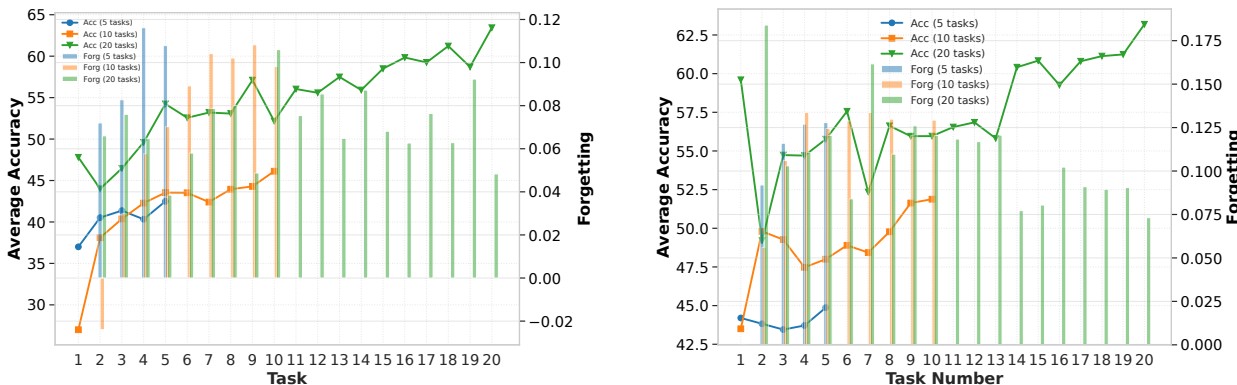

Figure 17: Scalability with increasing number of tasks on Split-TinyImageNet. **Left:** `COLD-ORACLE`. **Right:** `COLD`.

**Effect of varying** $\delta$**:** In `COLD-ORACLE` and `COLD`, the parameter $\delta$ controls the amount of forgetting that is tolerated. A larger $\delta$ allows greater degradation in performance on previous tasks, whereas a smaller $\delta$ imposes stricter constraints and limits forgetting. Figures 21 and 22 illustrate this behavior for `COLD-ORACLE` and `COLD`, respectively. In both figures, the top panel shows the evolution of the virtual queues, while the bottom panel reports task accuracies. As $\delta$ increases, the virtual queues decrease in magnitude, indicating fewer constraint violations. For `COLD`, the queue activation becomes concentrated along the diagonal, suggesting that constraints are primarily enforced for recent tasks due to the use of the previous model as a reference. In contrast, `COLD-ORACLE` distributes queue activation more uniformly across past tasks, reflecting the use of a stable best-model reference. Correspondingly, the accuracy plots show that larger $\delta$ improves current-task performance at the expense of past-task accuracy, indicating increased forgetting. This trend is consistent across both methods, highlighting the role of $\delta$ in controlling the stability–plasticity trade-off.

## G.5 Complexity and Scalability

Unlike projection-based methods such as GEM, which require solving a quadratic program at each update step with complexity scaling polynomially in the number of constraints, `COLD-ORACLE` performs a single gradient update with weighted replay gradients. Thus, its per-iteration computational complexity is equivalent to standard replay training, i.e., the complexity of `COLD-ORACLE` is comparable to techniques such as ER and `ER-ACE`. Furthermore, GEM requires constructing a gradient matrix of past task gradients at each step, `A-GEM` requires computing a full gradient over the memory buffer for projection, whereas COLD avoids both by using scalar queue weights and standard replay gradients, incurring only $\mathcal{O}(T)$ additional scalar storage. If

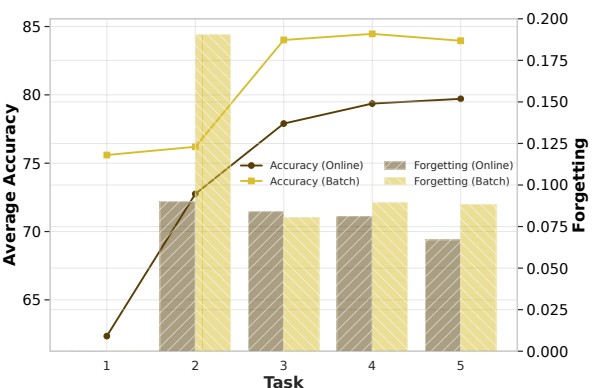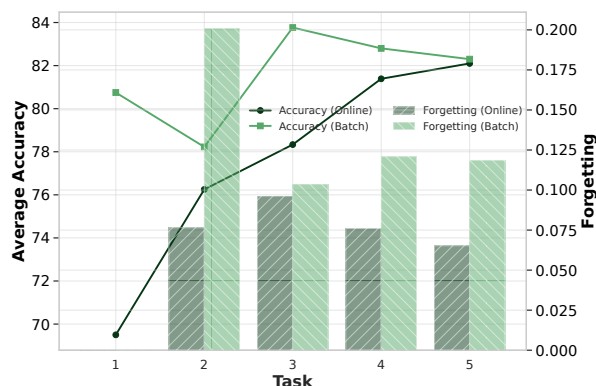

Figure 18: Comparison of online and batch settings on Split-CIFAR10. **Left:** `COLD-ORACLE`. **Right:** `COLD`.

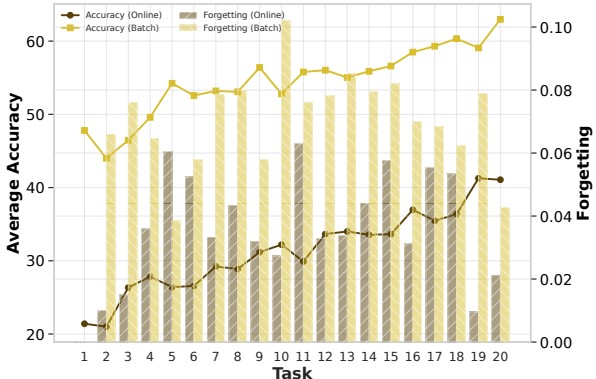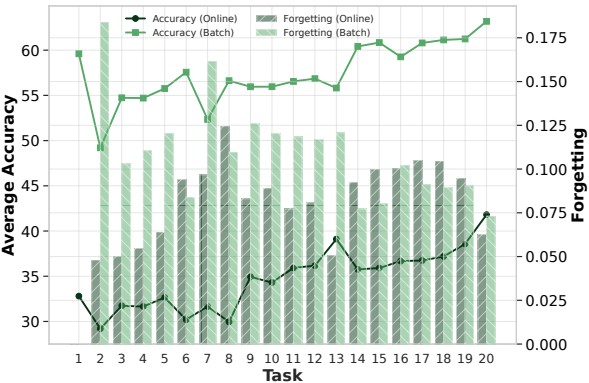

Figure 19: Comparison of online and batch settings on Split-TinyImageNet. **Left:** `COLD-ORACLE`. **Right:** `COLD`.

we compare the scalability of these methods, projection-based techniques shrink the feasibility regions as tasks accumulate, potentially degrading plasticity and increasing computational burden. In contrast, COLD regulates interference through scalar queue dynamics, avoiding geometric constraint shrinkage. Note that ER and `ER-ACE` simply mix replay gradients without constraint control; `COLD-ORACLE` matches the computational cost of ER while introducing only $\mathcal{O}(T)$ scalar queue variables to provide principled long-term stability guarantees.

# H    Complexity Analysis

In this section, we explicitly analyze the spatial and temporal complexity of our method and provide a comparative discussion with several baselines (ER-ACE, DER, DER++, GEM, A-GEM, EWC, MER, and NCCL).

**Notation:** We use the following notation to describe the computational complexity of the benchmark schemes:

- $t$: number of tasks seen so far,

- $m$: memory batch size per task,

- $M$: total memory buffer size,

- $C$: per-sample gradient computation cost,

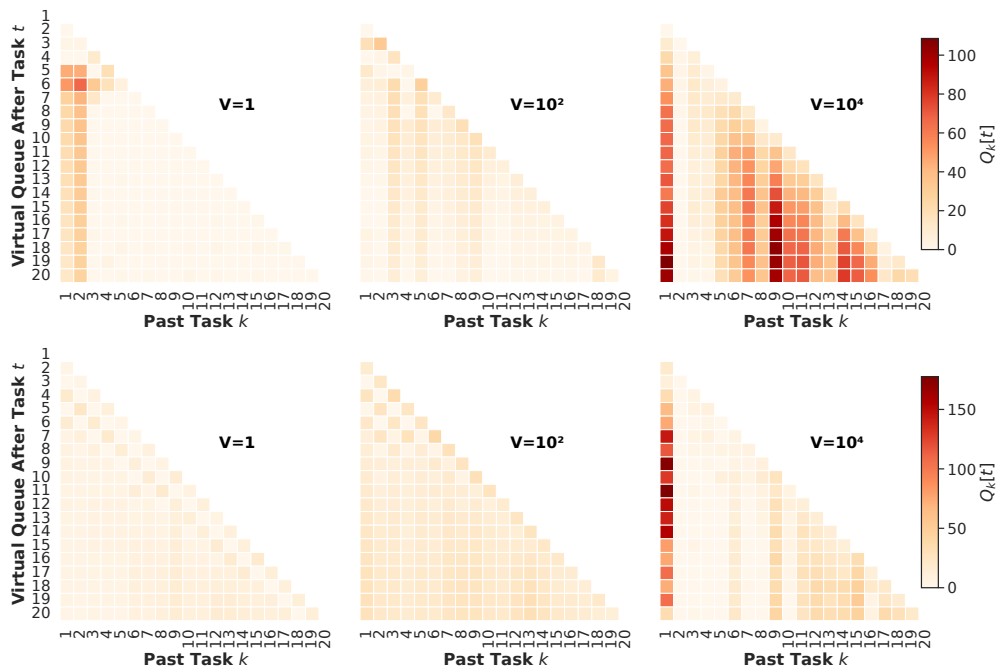

Figure 20: Virtual Queue comparison for different $V$ on Split-TinyImagenet dataset using `COLD-ORACLE` (Top) and `COLD` (Bottom).

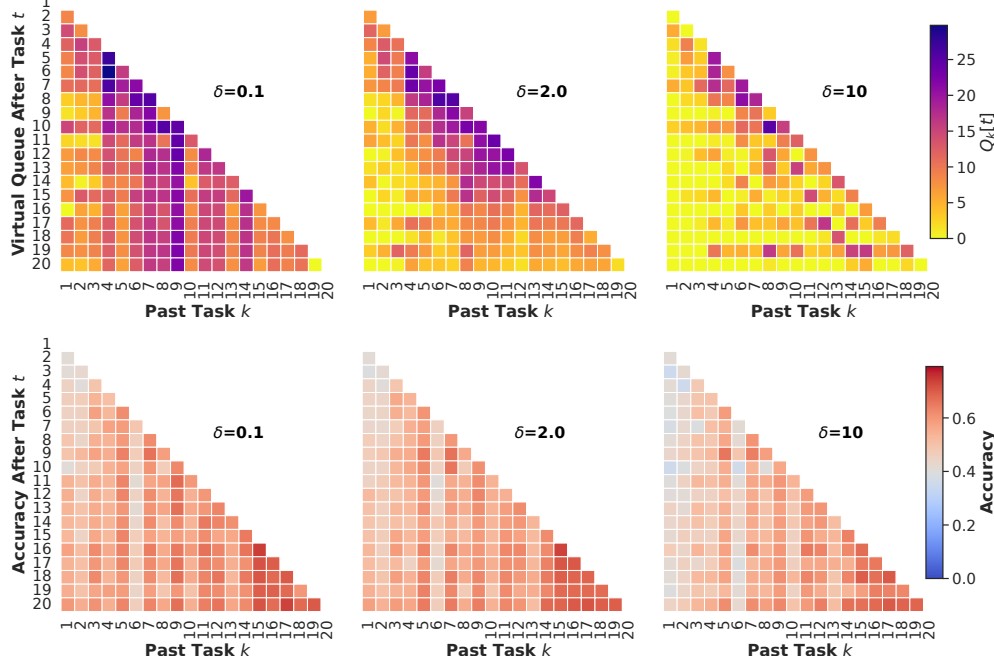

Figure 21: Effect of varying $\delta$ on `COLD-ORACLE` for Split-TinyImageNet. **Top:** Virtual queue evolution. **Bottom:** Task accuracy across tasks.

- $d$: model parameter dimension,

- $b$: current task batch size,

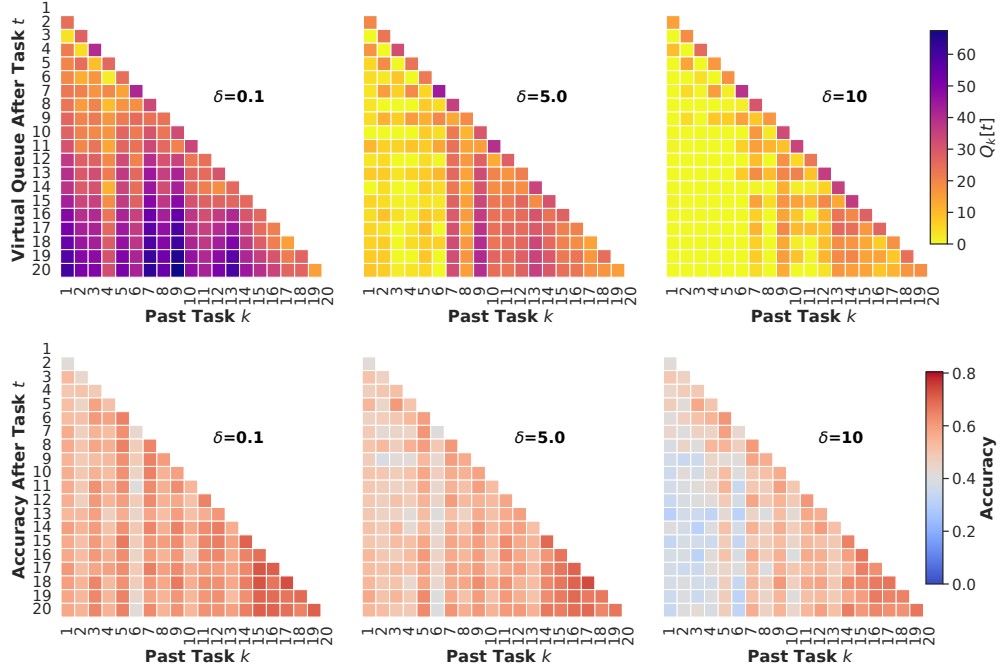

Figure 22: Effect of varying $\delta$ on `COLD` for Split-TinyImageNet. **Top:** Virtual queue evolution. **Bottom:** Task accuracy across tasks.

| Method | Per-batch Compute | Additional Storage | Constraint Mechanism |
|---|---|---|---|
| **Fine-tune** | $O(bC)$ | None | None |
| **EWC** | $O(bC + td)$ | $O(td)$ (Fisher diag.) | Parameter regularization |
| **A-GEM** | $O(mC + d)$ | $O(Mp)$ | Gradient projection |
| **GEM** | $O(tmC + t^2d)$ | $O(Mp)$ | Per-task QP projection |
| **MER** | $O(k(b + m)C)$ | $O(Mp)$ | Meta-gradient minimization |
| **GDumb** | $O(MCE)$ (inference) | $O(Mp)$ | Retraining at test time |
| **DER** | $O((b + m)C + mn_c)$ | $O(M(p + n_c))$ | Logit distillation |
| **DER++** | $O((b + m)C + mn_c)$ | $O(M(p + n_c))$ | Distillation + replay |
| **ER-ACE** | $O((b + 2m)C + bn_c)$ | $O(Mp)$ | Asymmetric replay loss |
| **CBA** | $O((b + 2m)C)$ | $O(Mp)$ | Bi-level optimization |
| **NCCL** | $O((b + m)C + d)$ | $O(Mp)$ | Adaptive learning rates |
| **REFRESH** | $O(k'(b + m)C)$ | $O(Mp)$ | Unlearn–relearn |
| **COLD** | $O((b + m)C)$ | $O(Mp + t)$ | Queue reweighting |
| **COLD** | $O((b + tm)C)$ | $O(Mp + td + t)$ | Queue reweighting |

Table 7: Comparison of computational and storage complexity across continual learning methods.

- $n_c$: number of classes,

- $p$: per-sample storage requirement,

- $k$: number of inner-loop steps (for meta-learning methods),

- $k' = k_u + k_r$: total unlearning and relearning steps,

- $E$: number of training epochs at test time for GDumb.

From the Table, we observe that the proposed COLD methods achieve a favorable trade-off between computational efficiency and storage requirements as compared to existing continual learning approaches. Classical regularization-based methods such as EWC incur additional $O(t \cdot d)$ computation and storage, which scales poorly with the number of tasks. Projection-based methods like GEM and A-GEM impose stronger constraints but at significantly higher computational cost, with GEM requiring quadratic programming of complexity $O(t^2 \cdot d)$, and both relying on memory buffers. Replay-based methods (ER, DER, DER++,

37

ER-ACE, CBA, NCCL) exhibit linear complexity in batch and memory size, but require additional storage for samples and, in some cases, logits. Furthermore, MER and REFRESH introduce further computational overhead due to meta-learning or iterative unlearning–relearning procedures.

