# OpenReview forum: "Theoretical Foundations of Continual Learning via Drift-Plus-Penalty"
_TMLR — Accepted by TMLR_

### Review · Reviewer_qgp6 · 2026-02-27

**Summary Of Contributions:**

# Short summary
This paper presents theoretical findings on Continual Learning (CL) by considering the problem as a long-term constrained stochastic control problem. Specifically, the paper adapts a Drift-Plus-Penalty principle to CL by framing the problem as a controlled process. Specifically, the problem is modeled as a sequence of constraints on previous tasks to limit forgetting and constrained on the current task to foster adaptation. One key element here is the usage of memory to estimate forgetting of the current model on previous tasks. The paper derives various theoretical guarantees, given a few assumptions, namely 1) smoothness of the loss 2) compactness of the parameter space, and 3) DPP can be solved exactly. It is important to note that this analysis and framework are applicable to memory methods only. While this is interesting, the community is shifting toward memory-free methods as much as possible, at least for offline scenarios like the one presented in the paper. Eventually, the paper shows the efficiency of the proposed methods compared to somewhat old baselines, in simple and well-studied datasets.

# Contributions
- The framework is well-motivated and interesting
- The theoretical framework is rigorous and well-presented
- The fact that the stability, plasticity, task-variation and memory quality are explicitly quantified in the framework is elegant
- The detailed experimental section shows clearly the impact of hyper-parameters
- The experiments show strong performances against well-known baselines, confirming the theory-guaranteed effectiveness

# Weaknesses

- **Absence of discussion on computation:**
While the theoretical guarantees are interesting, they rely on computationally costly requirements. One model must be stored for each task, and similarly, computing the update rule in equation (14) will linearly increase with the number of tasks. This impact spatial and temporal complexity of the method overall. Such an aspect should be detailed in the paper. Similarly, a comparison with other methods of computation is required to give a full, detailed study of the approach. In general, **potential limitations of the approach are barely discussed**.

- **The framework has guarantees on larger task sequences, but larger task sequences are especially costly in this framework:** The paper claims, p5, that "Average constraints prevent a single outlier task from dominating updates while still guaranteeing asymptotic feasibility via queue stability.". Firstly, I believe this to be an over-claim as means are sensitive to outliers, second, this is mostly true for longer sequences, and as developed above, longer sequences are increasingly costly. Also, focusing on the average favors stability over plasticity, as only the last task loss accounts for plasticity, while all previous task losses account for stability. Similarly, in p.11 "then larger number of tasks T ensures a smaller gradient average while keeping the queue bounded" displays guarantees for a larger number of tasks, which is again made unrealistic by the design of the framework. This should be fully discussed.

- The method is highly sensitive to $V$, which is a hyper-parameter. Introducing hyper-parameters in CL is always complex and may also hinder the realistic applications of the proposed approach for newer datasets. How to choose $V$ in a realistic case where hyper-parameters must not be searched for?

- In p9, $w_t$ is also estimated on memory data during training. Therefore, there might be a risk of over-fitting on those memory data, hindering the evaluation of $\phi^\hat(w_t)$. Would this invalidate the framework? Such an aspect is not discussed in the paper.

- in p2; the fact that the learning rate is fixed is presented as an advantage, but I do not understand why. Adaptive learning rate optimizers are usually more performant. Does this mean that the method works only on standard SGD, or could it be used in combination with a more evolved optimizer like Adam?

- In Figure 4: the accuracy of the presented methods varies drastically. However, for COLD, COLD-eff, DER, the first task should follow the same (or very close?) optimization process, according to equation 14 with $t=1$. **Why are the performances at the end of the first task different**? Also, why not include standard Experience Replay?

- The current method can only be applied to offline, well defined and memory based, CL problems

- Previous work mentioned and compared to is a bit outdated. However, the main contribution of the paper is theoretical, so I don't think this is a major problem.

- small over-claim: task order dependency problems are well-known in the literature, even though the theoretical insight given in this study is valuable.

## Typos and similar
- p7: Table 1 should be algo 1
- The text in some figures is hard to read

**Audience:**

Yes

**Audience Explanation:**

The theoretical findings are interesting and the framing can be valuable to bridge CL community and constrained stochastic control problem experts.

**Claims And Evidence:**

Yes

**Claims Explanation:**

I would answer yes as I believe it to be mostly the case given the width of experiments, ablation and sensitivity analysis. However I am concerned by the results of Figure 4 as I explained above. It seems that the performances should be the same or very similar after training on the first task for various methods. If the authors can clarify this point I would happily change my claim.

**This has been addressed by the authors, so I updated my review**

**Requested Changes:**

I would like to see:
- details on the computation
- discussion on theoretical and practical limitations of the approach
- clarifications on Figure 4

---

> ### Author Response · Authors · 2026-04-18
>
> **Regarding "Absence of discussion on computation:"**
>
> **Response:** We thank the reviewer for this important and constructive observation. We agree that storage and computation scaling with the number of tasks is an important aspect. A subtle point here is that such dependencies are not unique to our approach, and are in fact common across several established CL methods, including regularization-based methods, meta-learning based methods and replay-based methods. We acknowledge that this detail is not sufficiently described in the current manuscript. We will revise the manuscript to (i) explicitly analyze the spatial and temporal complexity of our method, and provide a comparative discussion with several baselines (ER-ACE, DER, DER++, GEM, A-GEM, EWC, MER and NCCL, to be included in the supplementary), and (ii) provide a paragraph in the manuscript clearly outlining the time and space complexity, and the limitations of our approach.
>
> **(i) Spatial and Temporal  Complexity and comparision** We provide a detailed comparision of complexity of our method and existing approaches as follows:
>
> **Notation:** We use the following notations to describe the computational complexity of various benchmark scheme:
>  - $t$ = number of tasks seen so far,
>  - $m$ = memory batch size per task,
>  -  $M$ = total memory buffer size,
>  - $C$ = per-sample gradient cost,
>  - $d$ = model parameter dimension,
>  - $b$ = current task batch size,
>  - $n_c$ = number of classes,
>  - $p$ = per- sample storage requierment
>  - $k$ = number of inner-loop steps (for meta-learning methods).
>  - $k'= k_u + k_r$(unlearning and relearning steps respectively)
>  - $E$ = number of training epochs at test time for GDumb
>
>
> | Method | Per-batch Compute | Additional Storage | Constraint Mechanism |
> |:---- |:---|:---|:---|
> | **Fine-tune** | $O(b  C)$ | None | None |
> | **EWC** (2017) | $O(b C + t  d)$ | $O(t  d)$ (Diagonal entries of Fisher matrix) | Per-parameter regularization |
> | **A-GEM** (2018) | $O(m  C +d)$ | $O(Mp)$ (Memory buffer) | Gradient projection (single reference) |
> | **GEM** (2017) | $O(t m  C)$ + $O(t^2d)(QP)$ | $O(M p)$ (Memory buffer) | Gradient projection (per-task QP) |
> | **MER** (2019) | $O(k  (b+m)  C)$ | $O(M  p)$(Memory buffer) | Meta-gradient interference minimization |
> | **GDumb** (2020) | $O(M  C  E)$(at inference) | $O(M  p)$ (Memory buffer) | Full retrain from scratch at test time |
> | **DER** (2020) | $O((b+m) C +m  n_c)$ (distillation) | $O(M  (p + n_c))$ buffer + logits | Logit-level knowledge distillation |
> | **DER++** (2020) | $O((b+m)  C)+ m  n_c)$ (logit terms) | $O(M  (p + n_c))$ (buffer + logits) | Logit distillation + label replay |
> | **ER-ACE** (2022) | $O((b+2m) \cdot C+b\cdot n_c)$ + (Asymmetric loss) | $O(M \cdot p)$ (Memory buffer) | Asymmetric cross-entropy on replay |
> | **CBA** (2023) | $O((b+2m) \cdot C)$ | $O(M \cdot p)$ | Bi-level optimization for online posterior distribution alignment |
> | **NCCL** (2023) | $O((b + m) \cdot C)$ + adaptive LR $O(d)$ | $O(M \cdot p)$ (Memory buffer) | Task-specific adaptive learning rates |
> | **REFRESH** (2024) | $O(k'  (b+m) \cdot C)$ unlearn+relearn passes | $O(M \cdot p)$ memory buffer | Iterative unlearn-then-relearn strategy |
> | **COLD-Eff** (Ours) | $O((m+b)  C)$ | $O(M  p + t)$ scalars | Virtual queue adaptive reweighting |
> | **COLD** (Ours) | $O((b+t m)  C)$ | $O(M  p+ t  d + t)$ scalars | Virtual queue adaptive reweighting |
>
> From the Table, we observe that the proposed COLD methods achieve a favorable trade-off between computational efficiency and storage requirements as compared to existing continual learning approaches. Classical regularization-based methods such as EWC incur additional $O(t  d)$ computation and storage, which scales poorly with the number of tasks. Projection-based methods like GEM and A-GEM impose stronger constraints but at significantly higher computational cost, with GEM requiring quadratic programming of complexity $O(t^2 d)$, and both relying on memory buffers. Replay-based methods (ER, DER, DER++, ER-ACE, CBA, NCCL) exhibit linear complexity in batch and memory size, but require additional storage for samples and, in some cases, logits. Furthermore, MER and REFRESH introduce further computational overhead due to meta-learning or iterative unlearning--relearning procedures.
>
> **(ii) Space and time complexity + Discussion of Limitations:** **COLD-Eff** maintains a computational complexity of $O((b+m)C)$, comparable to standard replay methods, while introducing only a negligible overhead of **$O(t)$ scalar storage** for virtual queues. The full **COLD** variant offers enhanced constraint expressivity by incorporating task-wise replay, at the cost of $O((b + t  m) C)$ computation and $O(t d)$ additional storage. Crucially, both variants replace expensive projection or meta-learning steps with a lightweight queue-based reweighting mechanism, enabling explicit control of forgetting with minimal computational overhead.

---

> ### Author Response · Authors · 2026-04-18
>
> We agree that the paper underemphasizes limitations. In the revision, we will add
>
> 1. **Computational complexity:** The computation complexity of gradient computation grows linearly with $T$ for COLD; this can be mitigated via task subsampling, windowed replay, or compressed model summaries.
> 2. **Choice of Trade-off parameter $V$:** The performance of the method depends on the choice of $𝑉$, which governs the plasticity–stability trade-off. While our analysis provides theoretical guidance, designing mechanisms to adapt $𝑉$ automatically during training remains an open problem. We will clarify this limitation and identify it as an important direction for future work.
> 3. **Fixed Learning Rate Assumption:** The current analysis assumes a fixed learning rate for simplicity, tractability and low algorithmic complexity. While this is standard in theoretical treatments, practical implementations often benefit from adaptive or scheduled learning rates. Extending the analysis to time-varying or adaptive learning rates is non-trivial, particularly due to their interaction with queue dynamics. We identify this as an important direction for future work.
>
> **Regarding "The framework has guarantees on larger task sequences":**
>
> **Response:** We thank the reviewer for the comment. We respectfully clarify the points below.
>
> **(1) On “mean is sensitive to outliers” and over-claim":** The reviewer is correct that sample means can be sensitive to outliers in general statistical settings. However, this concern does not directly apply to our formulation, for two reasons:
>
> The constraint in (3) is not a passive statistical average, but is actively controlled via virtual queue dynamics. The queue update as in equation (5) of the manuscript ensures that any persistent violation (including outliers) is accumulated and penalized in future updates. Consequently, an “outlier task” does not get ignored by averaging. Instead, it increases the corresponding queue, which increases its weight in subsequent optimization steps, thereby forcing corrective updates over time. This is fundamentally different from standard averaging that the mechanism is adaptive and feedback-driven, not static. Thus, the statement on page 5 should be interpreted in the following Lyapunov/DPP sense: "average constraints combined with queue stability ensure that no task can be persistently violated, even if it is an outlier."
>
> **(2) On "guarantees require long sequences which are costly":**  This concern seems to mix up theoretical guarantees (which are asymptotic in nature) with computational complexity (which governs per-iteration cost). The asymptotic guarantees (Theorems 1–2) are standard in stochastic control and online optimization, where performance is characterized in terms of time averages. Importantly, the per-iteration computational cost does not scale with sequence length in the way suggested: For COLD-Eff, the cost remains $\mathcal O(mC)$, independent of $t$. Even for COLD, the cost is comparable to existing baselines such as GEM, which already scale with $t$. Thus, longer sequences improve statistical averaging (as in any online method), but do not introduce additional per-step computational burden beyond standard CL baselines.
>
> **(3) On "average favors stability over plasticity":** This interpretation overlooks the core design of the DPP objective. The optimization objective is to solve
>
> $$\min_{\mathbf w} V \Phi_t(\mathbf w) + \sum_{k< t} Q_k[t-1] \Delta \hat \Phi_k(\mathbf w,\tilde{\mathbf w}_{t,k}).$$
>
> This explicitly separates plasticity governed by the current task loss scaled by $V$ with stability governed by the queue-weighted past constraints.
>
> *Important:* The parameter $𝑉$ provides a provable and tunable trade-off (Theorems 1–2):
> - Performance = $\mathcal O(1/V)$
> - Queue (stability/constraints) = $\mathcal O(V)$
>
> Larger $𝑉$ leads to more emphasis on plasticity while smaller $𝑉$ emphasizes on stability. Therefore, the framework does not inherently favor stability; it provides an explicit control knob to balance stability and plasticity. This is in contrast to GEM/A-GEM (hard constraints without theoretical guarantees), and adaptive replay methods such as NCCL (no explicit control of stability versus plasticity).

---

> > ### Author Response · Authors · 2026-04-18
> >
> > **Regarding "Risk of overfitting on memory data"**
> >
> > **Response:** We thank the reviewer for raising this important point. While the point is valid, we would like to clarify that the concern does not invalidate the framework. First of all, Memory replay is standard and not specific to our method. The use of memory data during training is inherent to replay-based CL methods such as ER, GEM, and A-GEM. Hence, any potential overfitting to memory is a general limitation of the setting. However, our analysis explicitly accounts for the memory approximation. For example, the theoretical guarantees explicitly include the deviation between empirical memory loss and true loss through the term $D_{\ell,\hat{\ell}}[T]$. This ensures that the framework remains valid even when the memory is only an approximate proxy. Secondly, control-based formulation mitigates overfitting. When the model performs well on memory data, the corresponding queue terms reduce, which automatically shifts more emphasis to the current task. This adaptive reweighting prevents excessive focus on memory and reduces the risk of overfitting. These observations are also supported by empirical evidence across multiple datasets and memory sizes. We see from Fig. 5 in the manuscript that the proposed method achieves strong performance, indicating that overfitting to memory does not hinder generalization in practice.
> >
> > **Regarding "Fixed learning rate versus adaptive"**
> >
> > **Response:** We agree with the reviewer that adaptive optimizers (e.g., Adam) along with adpative learning rates often provide better empirical performance. Our point about using a fixed learning rate is not intended as a practical advantage per se, but rather as a modeling choice that enables a clean theoretical analysis. In particular, the use of a fixed step size allows us to derive explicit convergence and stability guarantees, and clearly characterize the interaction between gradient updates and queue dynamics. Introducing adaptive learning rates would significantly complicate the analysis, as the effective update becomes state-dependent and intertwined with the queue evolution.
> >
> > Importantly, this restriction applies only to the analysis, not to the algorithm itself. The proposed method is fully compatible with standard optimizers such as Adam or other adaptive schemes, and can be implemented in conjunction with them in practice. We will clarify this distinction in the revised version. Lastly, despite using a fixed learning rate, the proposed algorithm still outperforms existing methods that are based on adaptive learning rates (eg: NCCL).

---

> ### Author Response · Authors · 2026-04-18
>
> **Regarding "Different initial performance"**
>
> **Response:** We thank the reviewer for this careful observation. This question raises two distinct points, which we address separately and in full detail below.
>
> - **Why do COLD, COLD-Eff, and DER show different first-task accuracy?**
> At $t=1$, the optimization objective is structurally identical across all replay-based methods, i.e., no past tasks, no memory replay, no queue penalty. The observed differences arise entirely from independently tuned hyperparameters. More specifically, the effective learning rates, the optimizers (ADAM vs. SGD-M), and batch sizes are different across methods (Tables 2–3 in supplementary), leading to different performance on the first task. It is important to note that the hyperparameters are rarely tuned to maximize first-task accuracy in isolation. Instead, techniques are tuned to maximize final average accuracy over all $T$ tasks. The first-task variation in Figure 4 is therefore a feature of independent long-run tuning, not a structural inconsistency.
>
> To fully address this concern, we provide controlled experiments where all methods use identical optimizers and learning rates. As shown in the table below, when all methods are trained with Adam ($\eta = 3\times10^{-4}$), first-task accuracies are nearly identical (within 1%), confirming that the differences in Figure 4 are purely attributable to hyperparameter choices and not to any algorithmic bias. Notably, COLD and COLD-Eff match ER exactly under both unified settings, consistent with the fact that all three reduce to the same objective at $t=1$. We additionally note that DER and DER++ employ online reservoir sampling, whereby the replay buffer is populated and queried within the first task itself. This introduces a mild self-regularization effect absent in other methods, which only activate memory replay from $t=2$ onward. This accounts for the accuracy gap observed for DER and DER++ even under unified hyperparameters.
>
> | Method | Tuned HP (Fig. 4) | Adam, $\eta =3 \times 10^{-4}$ | SGD, $\eta=3 \times 10^{-4}$ |
> |--------|---------|-------------------|------------------|
> | ER | 66.00 | 74.00 | 50.40 |
> | DER | 70.00 | 74.60 | 49.40 |
> | DER++ | 70.20 | 75.00 | 55.40 |
> | COLD-Eff | 73.60 | 74.00 | 50.40 |
> | COLD | 74.00 | 74.00 | 50.40 |
>
> **2. Why standard experience replay (ER) was not included in the main comparison?:** Our baseline selection aimed to cover a representative set of methods across all major CL families. ER-ACE is a direct and principled extension of ER that addresses a known weakness of standard ER representation drift under asymmetric updates and consistently outperforms standard ER on the same benchmarks. Including ER-ACE therefore subsumes ER as a weaker special case. Based on the well-established performance hierarchy in the CL literature where ER-ACE (Caccia et al.,2022), DER, and DER++ (Buzzega et al.,2020) all consistently outperform standard ER on Split-CIFAR10, Split-CIFAR100, and Split-TinyImageNet. We expect COLD and COLD-Eff to demonstrate an even more pronounced accuracy advantage over standard ER than over the stronger baselines already included. For the sake of completeness, we have included the following table to show the performance of ER.
> | **Method** | **CIFAR-10 (0.6k)** | **F** | **CIFAR-10 (1k)** | **F** | **CIFAR-100 (1k)** | **F** | **CIFAR-100 (5k)** | **F** | **Tiny-ImageNet (2k)** | **F** | **Tiny-ImageNet (5k)** | **F** |
> |:-----------|:------------------:|:----:|:-----------------:|:----:|:------------------:|:----:|:------------------:|:----:|:----------------------:|:----:|:----------------------:|:----:|
> | **ER (2019)** | **79.31** | **0.09** | **81.98** | **0.02** | **57.60** | **0.08** | **68.56** | **0.34** | **23.89** | **0.03** | **49.34** | **0.15** |
>
> **Regarding "Current method can only be applied to offline, well defined and memory based, CL problems""**
>
> **Response:** We agree with the reviewer on this matter. Like many other existing CL algorithms, we too need well defined memory for our algorithm to work. Proposing and analyzing the memory free CL methods is beyond the scope of this work, and will be investigated as a future direction.
>
> **Regarding "small over-claim: task order dependency problems"**
>
> **Response:** We thank the reviewer for this observation. We agree that task order dependency is a well recognized challenge in the CL literature, and we did not intend to claim otherwise. Our contribution is not the identification of task order dependency per se, but rather the first formal and algorithm-dependent quantification of how inter-task variability is captured through our task variation measure $D_Φ[T]$. This explicitly governs both the performance bound and the forgetting bound within a unified control-theoretic framework. We will update the manuscript accordingly.

---

> > ### Author Response · Authors · 2026-04-23
> >
> > **Response to "method is highly sensitive to $V$..."**:
> >
> > We clarify the concerns of the reviewer along three points:
> >
> > - **Hyperparameters are unavoidable in CL:** Most continual learning methods rely on multiple hyperparameters such as regularization weights (e.g. EWC, MAS, LwF) , multiple learning rate schedules (e.g. NCCL)) that must be jointly tuned. Similarly, our approach requires tuning of the parameter $V$ without adding any additional burden compared to the existing methods.
> >
> > - **Interpretability of hyper-parameter $V$:** The trade-off parameter $V$ is interpretable since it has a clear control-theoretic interpretation, explicitly governing the stability--plasticity trade-off via an $O(1/V)$ versus $O(V)$ relationship. As a result, $V$ can be selected based on application requirements rather than exhaustive hyperparameter search. We have clearly highlighted this in Fig. 5 (center and right).
> > - **Sensitivity of $V$:** The method is not highly sensitive to $V$ as evidenced in empirical results. From Figs. $1-5$, we see that performance varies smoothly and predictably with $V$, without abrupt degradation across a wide range of values. This indicates reasonable robustness. To further clarify this, we evaluate COLD and COLD-Eff on CIFAR-10 and CIFAR-100 across $V \in [0, 6 \times 10^3]$. We observe that accuracy varies smoothly, without any abrupt degradation, demonstrating that the method remains stable and only weakly sensitive to $V$ across a broad operating range.
> > - **Future scope:** The current formulation uses a fixed $V$, consistent with the theory, but it naturally admits extensions where $V$ is adapted based on task variation or queue dynamics. Exploring such adaptive control mechanisms is a promising direction for future work.
> >
> >
> > *Accuracies (%) of COLD-Eff and COLD versus $V$:*
> >
> > | Dataset & Memory / V | <span style="white-space: nowrap;">max–min [0, 6]</span> | <span style="white-space: nowrap;">max–min [6, 6×10]</span> | <span style="white-space: nowrap;">max–min [6×10, 6×10²]</span> | <span style="white-space: nowrap;">max–min [6×10², 6×10³]</span> |
> > |---------------------|----------------------------------------------------------|--------------------------------------------------------------|------------------------------------------------------------------|------------------------------------------------------------------
> > | **Split-CIFAR10 COLD-Eff** | 1.12 | 6.29 | 4.66 | 5.88 |
> > | **Split-CIFAR10 COLD** | 7.64 | 6.74 | 3.90 | 3.81 |
> > | **Split-CIFAR10 COLD-Eff** | 1.92 | 0.92 | 8.13 | 8.14 |
> > | **Split-CIFAR10 COLD** | 5.95 | 1.15 | 0.64 | 3.82 |
> > | **Split-TinyImageNet COLD-Eff** | 3.41 | 6.25 | 7.95 | 5.61 |
> > | **Split-TinyImageNet COLD** | 2.46 | 5.55 | 3.11 | 7.13 |

---

> > > ### Comment · Reviewer_qgp6 · 2026-05-07
> > > **Thank you**
> > >
> > > Thank you for your detailed replies which addressed my concerns. I will update my review accordingly.

---

> ### Author Response · Authors · 2026-05-10
>
> We thank the reviewer for accepting our clarifications and updating the review. If there are any further questions, we will be happy to answer.

---

### Review · Reviewer_87ip · 2026-03-24

**Summary Of Contributions:**

This paper proposes to tackle the stability-plasticity dilemma of CL by leveraging a control-theoretic approach. More specifically, it adapts the Drift-Plus-Penalty (DPP) principle to the CL setting, where the combined objective to minimise the loss on new tasks while not forgetting previous tasks is captured by a DPP-like objective. Under this view, the loss on the new task is seen as the penalty, while forgetting on previous tasks is measured by a per-task queue and is thus no longer treated as a static penalty but rather as a regulation of a dynamical system.

The main contributions are:

- casting CL under a control theoretic approach and adapting the DPP principle to the CL setting
- proposing COLD and COLD-Eff, two practical algorithms to solve CL under DPP
- providing theoretical guarantees on the proposed solution
- showing how COLD allows explicitly controlling the stability-plasticity trade-off via a hyperparameter V, and that this tradeoff is directly related to task similarity
- showing the effectiveness of the proposed algorithm in traditional CL benchmarks

**Additional Comments:**

The method could be better motivated. Although it is per se interesting to open up a new direction for tackling CL, it is not immediately clear to the reader why certain properties of the method should be beneficial. Just as an example, is it intuitive that dynamically controlling an averaged forgetting queue is a better than a static constraint?
The paper is very theory-focused, which makes it strong, but sometimes difficult to follow, adding some textual explanations to the theoretical parts could be helpful.

**Audience:**

Yes

**Audience Explanation:**

The paper is well thought-through and sound. The authors propose an interesting theoretical contribution, opening a new line of research within CL based on control theory and DPP.

**Broader Impact Concerns:**

No concerns.

**Claims And Evidence:**

Yes

**Claims Explanation:**

The paper is well thought-through and sound. The authors propose an interesting theoretical contribution, opening a new line of research within CL based on control theory and DPP.
The paper is clearly written, comprehensive and provides a wide array of theoretical derivations and arguments.
The proposed method has the advantage of providing an explicit handle on the stability-plasticity dilemma.
The experimental results show the validity of the method in the CL setting and support via a wide array of ablation experiments the theoretical arguments made.

**Requested Changes:**

On the method:

- Add, maybe at the end of the Introduction, a structure of the paper. At the moment it’s hard to understand how sections relate to each other. i.e. Say we start with idealised CL formulation in (2), then move to something that could be implemented in (3), then in 4 propose how to actually implement it.
- Can you add verbal descriptions of what each theory contribution means? (theorems, corollaries, main terms like
$ \Delta \hat{\phi_k}(w, \tilde w_{t,k})$ ...). Maybe adding a short bold description for each of those would help.
- In eq. (3): You introduce $\tilde{w}_{t,k}$. Can you be more explicit about what the subindices mean and be explicit that a full description comes later? (otherwise the reader is left wondering)
- Be more explicit on memory costs for COLD vs COLD-eff vs other methods? Do you need storing all models after learning all previous tasks?

On the results:

Plots are very hard to read and parse. Some suggestions:
- make figures larger
- make sure all x/y labels are present (e.g. see fig 3)
- add titles when necessary (e.g. fig 6 to highlight COLD vs COLD-eff)
- add more detailed captions that allow quickly understanding the figure (e.g. fig 1, high vs low not explained in figure, and also say that left highlights effect on plasticity while centre effect on forgetting)
- 7.1. not clear what the actual base task is?
- On the “canonical O(1/v)-O(v) tradeoff”. Can you help the reader by being more explicit? e.g. mention it’s the canonical tradeoff of standard DPP-based algorithms
- There is no description on how you chose hyperparameters for related methods. Can you add a short sentence? Did they get the same computational budget as your own method for choosing those?
- 7.2.2. conclusion: “the proposed methods put more emphasis on plasticity and forward transfer” Why is this stated? Does it not depend on the choice of V? Maybe an interesting experiment would be to show how different number of tasks lead to different optimal V values?
- 7.2.3: Can you add short explanations on:
- What explains the reduced gap between COLD and COLD-eff with large buffer size?
- What makes COLD-eff vs cold so unstable in fig 6?
- Fig 7 colormap is difficult to understand with 3 different colours, can you choose a gradient with 2 colours only?

Minor:

- Can you change “myopically” to something less negative when describing other methods?
- In the “sequel”: seems to imply a new paper is coming, maybe say “in the following sections”
- End of page 8: you refer to theorem 4 to analyse the three distinct factors - should be theorem 2?
- Queue stability analysis in 7.2.3: “This implies that using V = 10^2...”. The previous sentence does not seem to imply anything in that direction.

---

> ### Author Response · Authors · 2026-04-23
>
> **Response to the "Requested Changes: On the method":**
>
> - We thank the reviewer for the positive comments. We have made all structural changes suggested by the reviewer (comments $1$-$3$ above). We have added the description of each theorem, lemma, corollary and the mathematical terms. In particular, we will explain the benefits of the proposed approach, and how this can open up new directions in CL.
>
> - **Response to "Be more explicit on memory costs for COLD vs COLD-eff vs other methods?":** We thank the reviewer for this comment. We have provided a detailed comparision of complexity/storage by providing a table with description in the response to the reviewer **qgp6**, which we shall be adding in the supplementary. In addition, we shall add the following paragraph in the main manuscript to describe the time and space complexity of the proposed framework:
> **COLD-Eff** maintains a computational complexity of $O((b+m)\cdot C)$ (see response to **qgp6** for the notations), comparable to standard replay methods, while introducing only a negligible overhead of **$O(t)$ scalar storage** for virtual queues. The full **COLD** variant offers enhanced constraint expressivity by incorporating task-wise replay, at the cost of $O((b + t \cdot m)\cdot C)$ computation and $O(t \cdot d)$ additional storage. Crucially, both variants replace projection or meta-learning steps with a lightweight queue-based reweighting mechanism, enabling explicit control of forgetting with minimal computational overhead.
>
> - **Response to "Do you need storing all models after learning all previous tasks?":** We have proposed two algorithms namely COLD and COLD-Eff. COLD requires storing of all the previous models while COLD-Eff requires only the previous model.

---

> > ### Author Response · Authors · 2026-04-23
> >
> > **On the results:**
> >
> >  **Response to editorial changes to the plots":** We thank the reviewer for the comment. We will update the paper by incorporating all the suggested changes.
> >
> >  **Regarding "$\mathcal O(V)$-$\mathcal O(1/V)$-trade-off":** We agree that the trade-off can be stated more explicitly. In the revision, we will clarify that this is the standard trade-off arising in Lyapunov optimization / Drift-Plus-Penalty (DPP) frameworks, where:
> >
> > - $\mathcal O(1/V)$ characterizes the optimality gap (performance), and
> > - $\mathcal O(V)$ characterizes the queue size (constraint violation).
> >
> > This trade-off is well-established in the DPP literature, and our result shows that the proposed method inherits this canonical behavior in the continual learning setting. We will add a brief explanation and appropriate references to make this connection clearer to the reader.
> >
> > **Regarding "7.1: base tasks":** The “base task” is a common reference point that captures the core structure shared by all tasks, with each task deviating slightly from it. For instance, low task variation refers to a small perturbation around the base task (small variance $c_2$), leading to tasks that are very similar to the base task, and in high variation, each task drifts significantly away from the base task over time and hence, tasks become increasingly different from the base task.
> >
> > **Regarding "how you chose hyperparameters":** In the revision, we will include a brief description of how hyperparameters were selected for all baselines. Specifically, we adopt the hyperparameters reported in the original works when the experimental settings align with ours. When configurations are not directly applicable, we have tuned the methods over standard ranges using validation performance. We ensure fairness by allocating a comparable computational budget for hyperparameter search across all methods, including our own. This will be explicitly stated in the revised manuscript to improve transparency and reproducibility.
> >
> > **Regarding Conclusion: Plasticity and forward transfer:** We thank the reviewer for pointing this out. We agree that the statement in the conclusion may be misleading as written. The emphasis on plasticity versus forward transfer is not intrinsic, but is explicitly controlled by the parameter $V$. In our formulation, $V$ scales the current task loss and therefore directly controls the extent to which the model prioritizes learning new tasks (and hence forward transfer).
> >
> >
> > Our intent was to highlight that, compared to prior methods, the proposed framework enables stronger plasticity when desired, while still maintaining long-term guarantees. We will revise the conclusion to make the dependence on $V$ explicit.
> >
> > **Regarding reduced gap between COLD and COLD-eff:** The key difference between **COLD** and **COLD-Eff** lies in the choice of reference model and how past-task constraints are enforced. **COLD** uses the best historical model for each task, leading to a stricter, low-variance constraint, whereas **COLD-Eff** uses the previous model, resulting in a weaker, replay-based constraint. In **COLD-Eff**, the replay-based gradient satisfies
> >
> > $$\nabla \hat{\Phi}^{\text{replay}}(w_t)
> > = \nabla \Phi(w_t) + \epsilon_M,
> > \quad \mathbb{E}\|\epsilon_M\|^2 = O\left(\frac{1}{{M}}\right),$$
> >
> > which introduces estimation noise due to finite buffer size $M$. When $M$ is small, the error $\epsilon_M$ is large, so the replay loss is a noisy estimate of the true loss. In this regime, COLD's best-model reference provides a genuine advantage by enforcing tighter constraints through the virtual queue.
> >
> > As $M$ increases, $\epsilon_M \to 0$, meaning the replay-based estimate becomes accurate and the empirical loss closely matches the true past-task loss. Consequently, all past models become approximately equivalent, and the best historical model and the previous model impose nearly identical constraints in the queue update. Thus, the performance gap diminishes because the choice of reference model becomes indistinguishable when the buffer provides a faithful approximation of past task distributions.

---

> > > ### Author Response · Authors · 2026-04-23
> > >
> > > **Regarding "COLD-eff vs cold in fig 6?":** It is evident from Fig. 6 (COLD (Center) vs COLD-Eff(Right) that COLD-Eff shows more instability across tasks compared to COLD. The reason is explained next:
> > >
> > > - **COLD-Eff (Right)** uses the immediate preceding model, i.e., $w_{t-1}$. This means the constraint baseline shifts at every task, so the queue update in Eq. (5) is chasing a moving target. Any bad update at task $t$ directly poisons the reference for task $t+1$, causing error to propagate sequentially.
> > >
> > > - **COLD (Center)** uses the best historical model, which acts as a more stable anchor. It doesn't shift unless a genuinely better model is found, so the queue dynamics are less sensitive to individual task fluctuations.
> > >
> > > In short, COLD-Eff's incremental reference model creates a compounding instability as $T$ grows, since each task's constraint is relative to a reference that itself may have been poorly learned. COLD's global best-model reference absorbs these fluctuations, leading to smoother accuracy and forgetting curves at scale.
> > >
> > > **Response to Minor comments:** We thank for suggesting the changes. We will update the manuscript to take care of all the minor comments suggested above.
> > >
> > > **Response to "Additional Comments:"** We thank the reviewer for this important suggestion. We agree that the motivation, particularly the intuition behind the queue-based mechanism can be better explained. We will improve the exposition in the revised version.

---

> > > > ### Comment · Reviewer_87ip · 2026-04-24
> > > > **Feedback**
> > > >
> > > > I thank the authors for the updates and their answers. My concerns have been addressed.

---

> ### Comment · Reviewer_87ip · 2026-05-07
> **Additional concern regarding contributions**
>
> In light of the most recent additions by the authors, I have these additional comments:
>
> The current COLD requires `O(td)` storage to retain all past model checkpoints, this clearly violates the constraints of a CL setting and makes the comparison in Table 1 completely unfair. Although the results for COLD-Eff are good and the theoretical derivations are done in this setting, currently the framing of the paper makes it sound like COLD is the main contribution but also a valid CL alternative.
>
> To make the contribution clearer and avoid confusion, I request the following. First, COLD should be explicitly reframed as a theoretical upper bound on what the DPP framework can achieve under relaxed storage assumptions, rather than a practical CL method. To highlight the above, I also recommend renaming the two variants to something along the lines of:
>
> - **COLD-Eff** → **COLD**
> - **COLD** → **COLD-Oracle**
>
> This more honestly reflects that COLD-Eff is the legitimate CL contribution with a proper memory budget, more complete practical theory, and competitive empirical performance.
> Potentially, a per-task model baseline would have been beneficial to allow comparing COLD-Oracle to a very simple method that has the same storage budget (but not strictly necessary at this point).

---

> > ### Author Response · Authors · 2026-05-10
> >
> > We thank the reviewer for the constructive comments. We agree that COLD-Eff satisfies the CL memory constraints and hence, the naming suggested by the reviewer is appropriate. We shall make the suggested changes in the manuscript.

---

> > > ### Comment · Reviewer_87ip · 2026-05-26
> > >
> > > I thank the authors for the changes made and for the renaming. I would still argue against explicitly presenting COLD-ORACLE as a valid algorithm in the context of CL (as done in abstract, "We propose two algorithms...", or other places of the paper). Other than this, my raised concerns have been addressed.

---

### Decision · Action_Editor_trvS · 2026-06-03

**Recommendation:** Accept as is

**Audience:**

Yes

**Audience Explanation:**

The paper analyses the problem of interference, the core problem in continual learning.
The general topic and the specific results are of interest to TMLR's audience.

**Claims And Evidence:**

Yes

**Claims Explanation:**

This paper studies the stability-plasticity dilemma in continual learning by framing it as a control-theoretic problem. By adapting the Drift-Plus-Penalty principle, the authors treat learning new tasks as penalties to minimise and past-task forgetting as a dynamic system regulated by virtual queues, allowing explicit control over the trade-off.

Reviewers critiqued the paper's practicality and clarity. The main concern was that the algorithm unfairly violated memory limits by requiring storage of all past models. They also flagged the lack of explicit discussion on spatial and temporal scaling costs, high sensitivity to the trade-off hyperparameter, potential overfitting to the memory buffer, and poorly readable figures and theorems.
The authors addressed these concerns by restructuring the paper and adding transparent documentation. They renamed their practical, memory-efficient algorithm to COLD and reframed the memory-heavy version as a theoretical upper bound called COLD-Oracle. Furthermore, they added a detailed complexity comparison against several baselines, documented hyperparameter constraints in a new limitations section, explained the intuition behind their theorems in plain text, and completely overhauled the readability of their plots.

Overall, major concerns were addressed and the reviewers unanimously recommended acceptance.